# Design and Sensing Frameworks of Soft Octopus-Inspired Grippers Toward Artificial Intelligence

**DOI:** 10.3390/biomimetics10120813

**Published:** 2025-12-04

**Authors:** Seunghoon Choi, Junwon Jang, Junho Lee, Da Wan Kim

**Affiliations:** 1Department SKKU Advanced Institute of Nanotechnology (SAINT), Sungkyunkwan University (SKKU), 2066 Seobu-ro, Jangan-gu, Suwon-si 16419, Gyeonggi-do, Republic of Korea; gargoyle@g.skku.edu; 2Department of Electronic Engineering, Korea National University of Transportation, 50, Daehak-ro, Daesowon-myeon, Chungju-si 27469, Chungcheongbuk-do, Republic of Korea; wertt1027@a.ut.ac.kr (J.J.); wnsgh1916@a.ut.ac.kr (J.L.)

**Keywords:** soft robotics, octopus-inspired grippers, suction adhesion, multisensory integration, neuromorphic intelligence, real-world deployment

## Abstract

Soft robotics provides compliance, safe interaction, and adaptability that rigid systems cannot easily achieve. The octopus offers a powerful biological model, combining reversible suction adhesion, continuum arm motion, and reliable performance in wet environments. This review examines recent octopus-inspired soft grippers through three functional dimensions: structural and sensing devices, control strategies, and AI-driven applications. We summarize suction-cup geometries, tentacle-like actuators, and hybrid structures, together with optical, triboelectric, ionic, and deformation-based sensing modules for contact detection, force estimation, and material recognition. We then discuss control frameworks that regulate suction engagement, arm curvature, and feedback-based grasp adjustment. Finally, we outline AI-assisted and neuromorphic-oriented approaches that use event-driven sensing and distributed, spike-inspired processing to support adaptive and energy-conscious decision-making. By integrating developments across structure, sensing, control, and computation, this review describes how octopus-inspired grippers are advancing from morphology-focused designs toward perception-enabled and computation-aware robotic platforms.

## 1. Introduction

Soft robotics has rapidly evolved to address the inherent limitations of conventional rigid robotic systems. Traditional robotic platforms, although powerful in terms of force output and precise trajectory control, remain constrained by their intrinsic rigidity [1,2,3]. These characteristics impose fundamental limitations in environments where safety during human–robot interaction, compliance with fragile structures, and adaptability to unstructured or unpredictable conditions are required [4,5]. For instance, rigid manipulators frequently apply excessive concentrated forces, resulting in damage to delicate objects or safety concerns in collaborative scenarios [4]. Moreover, rigid architectures adapt poorly to complex terrains, uneven surfaces, and fluidic environments where continuous adjustment of contact interfaces is essential. These limitations constrain the use of rigid robots in domains such as minimally invasive surgery, wearable assistive technologies, field exploration, and disaster response [6,7]. In these domains, mechanical compliance, adaptive deformation, and self-protection against environmental perturbations are prerequisites for practical operation. Consequently, a transition toward soft robotic platforms, constructed from elastomers, hydrogels, and other deformable materials, has been actively pursued [1,8,9]. These systems provide intrinsic flexibility, enhanced energy absorption, and improved safety, enabling contact-rich interactions with uncertain and dynamic environments [1,2]. Against this background, the study of bioinspired adhesion and manipulation mechanisms has become a central focus in soft robotics research.

Among the diverse biological models explored for soft robotic inspiration, the octopus remains one of the most compelling and functionally versatile examples [10,11]. Octopus arms exhibit extraordinary dexterity through distributed muscular hydrostats, providing multi-degree-of-freedom actuation and control without reliance on rigid skeletal structures [10,11,12]. The suction cups distributed along the arm surface further extend functional capacity by enabling strong yet reversible adhesion [13]. These suckers operate effectively across dry and wet conditions, conforming to smooth, rough, or curved surfaces, and allow rapid cycles of attachment and detachment [14,15,16]. The combination of arm compliance and suction adhesion provides octopuses with robust manipulation strategies that integrate grasping, lifting, sealing, and tactile exploration. These features directly address challenges faced in robotic manipulation of irregular or fragile objects. Translating such biological functions into engineered systems has motivated the development of octopus-inspired grippers that emulate arm wrapping, suction-based adhesion, and sensory feedback [17,18]. The biomimetic value of this model can be distilled into several design axes: adhesion strength and reversibility, conformability to varied topologies, tolerance to contamination and wet environments, and efficiency of actuation [14,15,19]. These axes form the foundation for subsequent classification and performance evaluation in octopus-inspired robotic research [20].

The initial wave of octopus-inspired robotic grippers can be characterized as the structure-oriented design, defined primarily by structural imitation with minimal integration of sensing or computation. Research efforts have concentrated on replicating the geometry and material properties of suction cups to reproduce negative-pressure adhesion [15,21]. Fabricated from silicone elastomers and other compliant polymers, these devices demonstrated the feasibility of grasping on planar, curved, and even moist surfaces. While successful in validating the fundamental adhesive mechanism, these systems remained limited to basic contact establishment and binary detection of attachment. Sensors, when present, served largely to confirm the presence or absence of suction rather than providing detailed information about load distribution, object properties, or environmental conditions. Control strategies were rule-based and reactive, lacking contextual awareness. The significance of this design category lies in demonstrating that octopus-inspired structural motifs could be engineered to deliver reversible adhesion and mechanical compliance [21]. However, the absence of perceptual capability and adaptive decision-making underscored the need for progression beyond purely structural replication.

The perception-integrated design of octopus-inspired grippers extended beyond structural mimicry by incorporating multimodal sensing and data-driven control strategies [22,23,24]. In addition to suction geometry, these systems embedded pressure transducers, strain gauges, optical fibers, and flow sensors to monitor adhesion states and mechanical deformation in real time. By integrating multiple signals, grippers began to classify material properties such as stiffness, surface roughness, and curvature [23,25]. Machine learning and signal processing approaches were increasingly applied to extract features from sensor outputs, enabling improved recognition of grasped objects and adaptive adjustment of suction parameters or arm positioning [24,26]. This development marked a transition from purely reactive adhesion to perceptive manipulation. Despite these advances, critical challenges persisted. Data-driven models required large volumes of training data and often lacked robustness when confronted with novel objects or unstructured environments [27]. Energy and computational demands also imposed practical constraints for mobile or wearable platforms [22]. Nevertheless, the perception-integrated approach established the foundation for perception-enabled gripping and highlighted the promise of multimodal sensing as a pathway toward intelligent manipulation.

Recent research has increasingly shifted from conventional sensing–actuation pipelines toward AI-centered integration, in which sensing, computation, and control are co-designed to operate as a unified system [27,28,29,30]. Instead of treating sensors as passive modules, these approaches embed computational functions directly at or near the sensory interface, enabling event-driven processing, multimodal fusion, and real-time inference with minimal latency [27,31,32]. This paradigm—often inspired by neuromorphic principles but implemented through modern AI frameworks—aims to achieve adaptive, context-aware behavior that remains robust even when encountering previously unseen objects, materials, or environments [27,28,33]. Emerging prototypes demonstrate the breadth of this direction: innervated suction cups capable of high-bandwidth deformation sensing, triboelectric tactile systems coupled with deep-learning classifiers, and fluidic logic architectures that emulate distributed decision-making seen in biological arms [22,23,26,34,35]. Although these demonstrations showcase clear potential, they remain at an early stage, and fully integrated intelligent grippers ready for field deployment have yet to materialize. Bridging this gap will require convergence across soft materials, multimodal sensing, flexible electronics, and AI-driven computational design.

In this review, octopus-inspired grippers are reorganized within a function-oriented framework that reflects the field’s shift toward tighter integration of structure, sensing, computation, and control. Systems are grouped according to their primary roles, including suction mechanisms, arm-type manipulators, hybrid assemblies, and sensor-focused configurations, enabling a clear comparison of design choices and their impact on adhesion reliability, deformation behavior, and environmental adaptability [15,17,18,36,37,38,39,40]. The discussion highlights how structural geometry, multimodal sensing, and the physical arrangement of sensors and actuators influence contact stability and manipulation capability [14,19,22,23,34,41,42,43,44]. Particular attention is given to recent advances in AI-assisted perception, which support alignment estimation, material identification, pose inference, and environmental classification through data-driven analysis of tactile, pressure, optical, and electromechanical signals [26,27,28,31,33]. These approaches move soft grippers beyond reactive motion and toward context-aware behavior, offering a foundation for adaptive and energy-efficient interaction inspired by neuromorphic principles while maintaining grounding in practical machine-learning models [27,28]. By synthesizing developments across materials, sensing architectures, and computational strategies, this review provides a consolidated basis for interpreting current capabilities and outlines research directions for deploying intelligent soft grippers in complex real-world environments.

This review provides representative studies on octopus-inspired suction and arm-based grippers published over the past decade, with earlier biologically foundational works included when relevant. The surveyed literature spans structural designs, sensing modalities, multimodal perception, and computational approaches, forming the functional categories used to organize this review.

## 2. Overview of Octopus-Inspired Soft Gripper Designs

Research on octopus-inspired grippers has advanced in diverse directions, reflecting differences in structural architectures, functional goals, and sensing modalities [17,36,37,45,46,47]. For systematic analysis, existing grippers can be classified along two primary axes: structural design and sensing capability [28,29,48,49]. The structural axis includes arm-type grippers, suction-based systems, hybrid mechanisms combining both, and specialized micro/nanostructured designs inspired by octopus suckers [15,17,38,45,47]. The sensing axis encompasses multiple modalities—pressure and force detection, strain and deformation measurement, temperature sensing, tactile and surface-texture recognition, and integrated multimodal systems that fuse these signals into coherent perception strategies [22,23,28,34,43,44,50]. This dual-axis framework establishes a foundation for comparing how different design strategies address key challenges, including adhesion reliability, structural conformability, energy efficiency, and environmental adaptability [9,14,19,41,42]. Table 1 summarizes representative studies in terms of material choice, environmental compatibility, actuation mechanism, geometric scale, and maximum adhesion/grasping force [14,19,26,43,44,46]. The data provide a structural and mechanical comparison supporting the suction/arm design analysis in Chapter 3. Moreover, the integration of sensing into grippers has progressed from simple contact confirmation to sophisticated multimodal perception and adaptive feedback [22,23,28]. Strain and deformation sensors, often based on conductive polymers, carbon nanotubes, or graphene composites, measure bending, stretching, and displacement of soft structures (Figure 1d(i)) [22,24,36]. Pressure and force sensors provide fundamental measurements of suction states, enabling estimation of contact quality and load distribution (Figure 1d(ii)) [21,50]. These allow reconstruction of gripper shape and monitoring of mechanical interaction with objects. Temperature sensors are incorporated in some designs to protect sensitive objects or to enhance safety in biomedical contexts [9]. Tactile and surface roughness sensors quantify frictional interactions and classify materials based on surface textures (Figure 1d(iii)) [32,43,44]. The most advanced implementations combine multiple modalities into integrated multimodal sensing systems [23,28,34,49]. By fusing heterogeneous signals, these systems can achieve object recognition, infer stiffness, estimate weight, and classify environmental states [26,28,31,33]. When coupled with machine learning, multimodal sensing enables adaptive control strategies that adjust suction levels, arm trajectories, or contact timing based on real-time feedback [23,26,31,33]. This evolution reflects a broader trajectory from structural mimicry toward intelligent perception [27,28,49].

**Table 1 biomimetics-10-00813-t001:** Quantitative comparison of octopus-inspired soft grippers based on material, actuation mechanism, and lifting performance.

Materials	Lifting Ratio [Max Force/Gripper Size, kPa]	Max Force [N]	Gripper Size (mm^2^)	Environment Condition	Actuation Method	Reference
Ecoflex/Dragon Skin	15.1	2	132.7	Dry	Pneumatic	[21]
Ecoflex/Dragon Skin	31.2	9.8	314	wet surface	Passive suction	[15]
Ecoflex	21.7	3.84	177	Dry	Pneumatic/Electric	[53]
Mord Star/Dragon Skin	40.6	26.99	665	Dry	Pneumatic	[18]
PDMS/Carbonyl iron particles	8.28	7.45	900	Dry/Underwater	Magnetic	[29]
Ecoflex/Dragon Skin	2.2	4.3	1963	Underwater	Pneumatic	[54]
MoldMax	8.3	3.43	415	Dry/wet surface	Pneumatic	[25]
PDMS/Dragon Skin	33.2	9.4	283	Dry/Underwater	Pneumatic	[23]
Agilus30Clear	9.2	9.35	1017	Dry	Pneumatic	[34]
Dragon Skin	56.0	45	804	Wet surface/Underwater	Pneumatic	[22]
Dragon Skin	3.3	3.8	~1154	Underwater	Pneumatic	[39]
Ecoflex/Dragon Skin	8.7	74.14	~8494	Dry/Underwater	Pneumatic	[52]

The organization of structural and sensing categories is further illustrated through visual schematics (Figure 1). These representations demonstrate how arm-based grippers mimic the wrapping and enveloping motions of octopus arms [17,36,47], how suction-type systems replicate the pressure-driven adhesion of suckers (Figure 1a(i–iii)) [14,15,19,46], how hybrid systems combine the advantages of both (Figure 1b(i–iii)) [17,38,39,40], and how special structures incorporate microstructural or chemical features for enhanced adhesion or sensing (Figure 1c(i–iii)) [14,43,44]. On the sensing side, schematics outline how each modality contributes specific information about interaction forces, surface states, or object properties (Figure 1d(i–iii)) [22,23,28,32,34]. Together, these frameworks provide the reference points through which existing studies can be compared and contextualized. Table 2 including strain, flow, pressure, optical, triboelectric, and multimodal systems are compared by sensing purpose, available grasping/adhesion force ranges, and ML-based classification or estimation performance [22,23,26,28,31,32,33,34]. This table establishes the sensing–perception landscape for subsequent sections on sensor-integrated manipulation and AI-driven control.

Suction-type grippers directly emulate the suction cups of octopuses, which achieve adhesion through negative pressure and frictional sealing (Figure 1a(i,ii)) [14,19]. Artificial implementations employ diverse actuation strategies, including pneumatic vacuum creation, electrically driven pumps, and liquid-mediated adhesion [41,46,50]. The central principle remains the establishment of a pressure differential that anchors the gripper to the target surface. Such devices are particularly effective in wet environments and on rough or curved surfaces [43,44], where traditional rigid suction mechanisms lose efficiency. They exhibit strong adhesion across challenging surface conditions and enable rapid attach–detach cycles [14,15,19]. However, these systems demand significant energy to maintain pressure differentials, and fabrication complexity increases notably when scaling to multi-unit arrays (Figure 1a(iii)) [17,38]. Synchronization of these arrays remains a technical challenge. Typical applications include medical suction patches that adhere to skin or tissues, as well as underwater exploration robots requiring reliable attachment in fluid environments [14,19,40,46]. The suction-type grippers show a broad range of sensor integration. For instance, flow sensors, as reported by Yue et al., are used to detect sealing quality and leakage in real time, supporting Contact/Adhesion, and Force/Load [51]. Strain gauges embedded in suction walls have been utilized for dual purposes—state estimation and environmental sensing—as demonstrated by Lee et al. [54], and Kim et al. [22]. Optical sensing expands capabilities further, with studies by van Veggel et al. illustrating how optical feedback can be used, classifying surfaces and monitoring adhesion interfaces [34]. These examples position suction systems as versatile platforms that merge adhesion with perception.

**Table 2 biomimetics-10-00813-t002:** Sensing strategies, functional roles, and machine-learning performance in octopus-inspired soft grippers. (A) State & Pose Estimation; (B) Contact/Adhesion & Force/Load; (C) Object/Material & Environment.

Type	Sensor	Purpose	Grasping Force	ML Performance	Reference
Suction	Flow velocity	(B) Contact/Adhesion & Force/Load	Seal: 1.5 kPa difference → up to ~0.35 kg	Recognition accuracy: 91.7%	[25]
	Strain gauge	(A) State & Pose Estimation (Proprioception & Alignment) (C) Object/Material & Environment	Dry 8.7 N, Wet 9.4 N	Weight error: ±8.63 g, COG error: ≤5 mm	[23]
	Strain gauge	(B) Contact/Adhesion & Force/Load	Dry 26 N, Wet 45 N	–	[22]
	Strain gauge	(A) State & Pose Estimation (B) Contact/Adhesion & Force/Load	Max 31.2 N	Shape accuracy: 98.3%, Hardness: 98.0%	[54]
	optical	(B) Contact/Adhesion & Force/Load (C) Object/Material & Environment	Max load ≈ 2 N	–	[21]
	optical	(B) Contact/Adhesion & Force/Load	Shear pull-off: 5.28 N	Orientation MAE: 1.97°/9.41°	[34]
Arm (single)	Strain gauge	(A) State & Pose Estimation (Proprioception & Alignment)	Max payload: 160 g	–	[24]
Arm (array)	Flow velocity	(B) Contact/Adhesion & Force/Load	Max payload: ~400 g	–	[40]
	pressure	(B) Contact/Adhesion & Force/Load	Pull-off: 0–9.8 N	Classification accuracy: 96.2%	[51]
	triboelectric	(B) Contact/Adhesion & Force/Load (C) Object/Material & Environment	Grasping force: ~2 N	–	[26].

Arm-type grippers are inspired by the muscular hydrostat structure of octopus arms, which allows continuous bending, twisting, and wrapping around objects without rigid skeletal components (Figure 1b(i)) [36,37]. In engineered systems, this capability is reproduced using elastomers, hydrogels, and other high-elasticity polymers that allow large deformations while maintaining structural integrity [9]. Such designs excel in grasping irregularly shaped or large objects by conforming their structure around the target. They often exhibit relatively simple architectures, relying on pneumatic or tendon-driven actuation to achieve bending motions [36,47]. The advantages of this class include environmental adaptability and the ability to secure fragile objects through distributed contact forces. However, their limitations include difficulty in manipulating small objects with high precision and insufficient adhesion when only structural wrapping is employed. These characteristics position arm-type grippers as effective for applications such as handling large components in industrial contexts or manipulating natural specimens in exploration scenarios. The Arm-type grippers also have mainly used strain gauges for proprioception. For instance, Xie et al. presented a single-arm design capable of estimating bending and alignment [24]. More advanced arm arrays introduced additional modalities: Wu et al. integrated flow sensors for distributed force detection [39], while Sareh et al. employed pressure sensors to monitor suction-cup dynamics, enabling multimodal perception of contact, medium changes, surface roughness, and pulling force [21]. A distinctive approach was described by Chen et al., where triboelectric sensors provided both contact-force measurement and environmental recognition [26]. Together, these studies show a progression from basic proprioception to multimodal perception in arm-type systems.

Hybrid grippers combining multiple suction modules with compliant arm segments integrate the advantages of both mechanisms. They further extend beyond macro-scale replication by incorporating microstructural and chemical enhancements inspired by octopus suckers (Figure 1c(i–iii)) [15,43,44]. The design leverages the synergistic interplay between flexible arm wrapping and suction anchoring, allowing stable grasping of diverse object geometries and environmental conditions. The arm (array) designs combine these strategies, embedding suction sensors in flexible arms to monitor both global deformation and local adhesion [17,34]. While more complex to control, they illustrate the convergence of structural compliance and suction sensing. Special-structure grippers often employ unconventional modalities; the triboelectric array by Chen et al. exemplifies self-powered sensing, simultaneously harvesting energy and recognizing surface interactions [26]. Such multifunctional designs point toward coupling adhesion and perception within a single interface.

At the macro level, hybrid arrays integrate multiple soft arms with embedded suction cups along their surfaces or tips [17,18,39]. The arm segments provide conformability and distributed contact forces, while localized suction modules ensure firm adhesion. Such systems excel at handling irregular, fragile, or slippery objects by combining continuous arm deformation with discrete suction events. However, their control complexity increases, as coordination is required between arm kinematics and suction activation. Typical implementations involve pneumatic or tendon-driven bending motions coupled with vacuum actuation, providing versatility in industrial manipulation and underwater exploration [36,47].

At the microstructural level, hybrid designs incorporate hierarchical ridges, fine surface hairs, or chemical coatings inspired by biological suckers [14,19,43,44]. These features enhance adhesion in wet or contaminated environments and sometimes couple adhesive performance with sensing functionality. Examples include multifunctional patches that adhere securely while simultaneously measuring strain, load distribution, or surface conditions [22,23]. Such features broaden environmental compatibility and enable potential applications in biomedical patches, surgical tools, and wearable devices.

The advantages of this category lie in redundancy and multifunctionality: distributed arms offer robustness against single-point failure, while microstructured suction interfaces improve performance across challenging environments [40,43,44]. Nonetheless, scalability remains a challenge, as fabricating uniform arrays with consistent suction and microstructural fidelity is technically demanding [38]. Despite these difficulties, multiple suction + arm grippers represent the most integrative approach, merging mechanical compliance, adhesion strength, and multifunctional sensing into unified platforms [17,18,39].

In summary, early systems were limited to structural replication with strain sensing, while later work—such as triboelectric (Chen et al.) [26] or optical (van Veggel et al.) [34] grippers—combined adhesion with perception. Arm-type systems laid the groundwork for proprioception [24,36], suction systems expanded versatility [15,43,44], hybrid designs added adaptability [17,18,39], and special-structure grippers introduced multifunctionality [14,19,22,23,34]. Together, these studies outline how octopus-inspired grippers diversified across structure and sensing, demonstrating a clear trajectory of progressive development.

## 3. Structural and Sensor-Integrated Modules

This section describes the framework of octopus-inspired grippers in terms of how structure, sensing, and computation have been increasingly integrated. While prior studies have often emphasized isolated achievements in design or sensing, a functional reorganization—focusing on suction mechanisms, embedded sensing, and arm/hybrid structures—allows us to clarify how these mechanical modules collectively support adaptive grasping performance [1,2,49].

Rather than interpreting these developments, we categorize existing designs according to their functional roles. At the first stage, structure-oriented designs focus on whether the system can perform stable grasping. These structural principles form the mechanical basis upon which sensing can be integrated to monitor seal formation, loading direction, or surface conformity [1]. Building on these structural elements, sensor-integrated suction designs incorporate multimodal sensing such as strain, optical, or pressure-based feedback to infer contact conditions and object-related properties [22,23,28]. These embedded sensing modules do not yet perform high-level decision-making; rather, they serve as physically grounded perception units that enhance the reliability of suction and improve the gripper’s ability to interact with irregular environments. Arm- and hybrid-type modules further extend these capabilities by providing continuum deformation, wrapping, and multi-surface adaptability [17,36,37]. Tentacle-inspired actuators integrate stretch or curvature sensors to achieve proprioception, distinguishing actuator-driven motion from externally induced deformation [24]. Wide-area suction disks and triboelectric or flow-based sensing structures also fall within this category, as they combine structural compliance with spatially distributed sensing to support more versatile grasping scenarios [26,39].

Together, these structural and sensor-integrated modules constitute the physical backbone of octopus-inspired soft grippers. Rather than presenting a stepwise evolution from structure to sensing to computation, this chapter examines each module according to its mechanical contribution, sensing capability, and integration strategy, providing the foundation for the operation- and AI-driven behaviors discussed in Chapter 4. Representative examples are summarized in Figure 2, Figure 3 and Figure 4.

### 3.1. Suction and Sensor-Integrated Suction Modules

Suction-based designs form one of the fundamental structural modules in octopus-inspired soft grippers, providing the primary mechanism for attachment, load transfer, and surface adaptation. These systems typically draw inspiration from the biological sucker’s infundibulum and acetabulum, incorporating geometric features such as stiffness gradients, tapered cavities, or multi-chamber air pathways to enhance conformal sealing on both smooth and irregular surfaces [13,15]. Rather than representing an early “stage” of development, these suction architectures serve as the core mechanical foundation upon which additional sensing and control capabilities can be integrated.

Structural imitation alone enables stable adhesion and consistent grasping performance across various target objects, as demonstrated in several engineering implementations [14,19,43,44]. Complementing these designs, research has explored the incorporation of basic deformation sensing—such as strain or pressure measurement—directly within the suction body to monitor seal formation, attachment quality, and loading direction [22,23]. These embedded sensing features provide real-time physical cues about object–gripper interaction without requiring external instrumentation. Together, these suction and sensor-integrated suction modules constitute the first functional building block of octopus-inspired grippers, forming the basis for the more complex arm and hybrid architectures discussed in the following section.

The designs in Figure 2 highlight three distinct structural strategies for embedding sensing directly into suction architectures: stiffness-gradient deformation measurement using optical fiber elongation, friction-enhanced suction with strain-based directional feedback, and multi-chamber pressure-field sensing for flow-based tactile inference [21,25,54]. The designs encapsulate rich structural–sensing mechanisms that offer complementary pathways for improving seal stability, detecting contact asymmetry, and inferring local surface geometry. Figure 2a illustrates the stiffness-gradient suction cup developed by Sareh et al., in which the infundibulum was engineered to deform in a controlled manner under negative pressure [21]. According to the schematic and cross-sectional images, the upper region of the infundibulum was fabricated to be more compliant, enabling it to collapse inward more easily under negative pressure, while the lower region remained stiffer to preserve dome geometry. This stiffness differential creates a controlled anchoring deformation that increases the effective sealing area with increasing vacuum. The embedded optical fiber, positioned centrally along the anchoring column, functions as a deformation transducer: as the infundibulum collapses inward under suction, the anchor length elongates, causing measurable changes in light intensity transmitted through the fiber. This produces a quantitative mapping between internal suction force and deformation magnitude. In Figure 2a(ii), the experimentally measured holding force exhibits a nonlinear increase from ~1.2 N to ~2.2 N over a pressure range of –0.1 to –0.5 bar. The finite element model, which incorporates elastic material behavior and axisymmetric suction loading, reproduces this trend closely, validating that the stiffness gradient governs the force–deformation response. As shown in Figure 2a(ii), The infundibulum undergoes a distinctive buckling–anchoring transition at higher negative pressures, and that the optical fiber signal reflects this transition with high temporal precision. This makes the sensor not merely a displacement indicator but a tool for characterizing the mechanics of seal formation and detachment as key parameters for suction-cup reliability.

In Figure 2b(i), the hierarchical gripper by Wu et al. integrates a suction cup with a hexagonal friction interface molded from a patterned elastomer [54]. Each hexagonal unit features micro-scale protrusions that engage with textured or slanted surfaces, increasing real-surface contact and resisting shear slip. Surrounding this patterned interface, strain gauges are laminated at orthogonal positions so that any asymmetric deformation which caused by tilt, lateral shift, or object rotation produces a distinct electrical response on one side relative to the other. This design allows the gripper to convert mechanical misalignment into measurable strain distributions. For example, when the cup contacts a surface at a slight angle (<10°), the strain gauge closest to the elevated edge registers a delayed or reduced strain peak relative to the opposing gauge. This directional asymmetry, clearly shown in the sensor plots, provides a basis for detecting object orientation before full suction is applied. As illustrated in Figure 2b(ii), the suction force over time shows the characteristic sequence of preload, attachment, steady holding (~4.3 N for a 430 g payload), and rapid detachment. The strain signals not only confirm the suction state but also track object-induced shear forces during dynamic motion such as shaking or lifting. This coupling between friction-enhanced stability and deformation-based sensing makes the structure robust to uneven surfaces and introduces an early form of “contact-aware adjustment,” where the gripper can infer misalignment and compensate for it prior to object lifting. The core mechanics including hierarchical friction, asymmetric strain sensing, and stable suction holding are fully retained.

The design in Figure 2c(i) employs a multi-chamber architecture, where the suction cup is partitioned into four internal air pathways connected to independent pressure sensors [25]. Each chamber acts as a localized flow channel that responds differently depending on where leakage occurs along the rim. When the suction cup seals imperfectly, tangential airflow enters through the gap, preferentially disturbing pressure in the nearest chamber. This creates a distinct pressure vector whose direction corresponds to the leakage location. The Computational Fluid Dynamics (CFD) results shown in Figure 2c(ii) demonstrate how slight differences in inflow velocity—on the order of 0.85–85.6 cm/s—produce quantifiable kPa-level gradients among chambers. These gradients scale with surface roughness and misalignment, enabling the system to estimate both the direction and magnitude of seal failure. Even small angular offsets (<15°) generate reproducible differential pressure signatures, forming the basis for leak localization. Functionally, this sensing method enables the cup to perform haptic search behaviors: if one chamber detects a significantly lower pressure, the system interprets this as a weak seal on that side and tilts the gripper to re-establish a uniform pressure distribution. The presented imagery captures the essential mechanism that distributed multi-chamber pressure sensing enables local geometry inference, seal-quality assessment, and orientation correction during contact.

The suction structures in Figure 2 form the mechanical basis for stable attachment, providing fundamental information such as seal quality, load transfer, and leak direction through structural deformation or localized pressure sensing. However, these responses are typically low-dimensional and primarily reflect global suction behavior. The designs in Figure 3 build upon this foundation by embedding more advanced sensing elements such as strain sensors, optical markers, and CNT-based tactile layers directly into the suction body [22,23,34]. These integrated modules enable the gripper to measure complex, spatially varying deformation patterns, allowing more precise interpretation of contact conditions. As shown in Figure 3a(i), a strain-sensor–integrated suction cup is fabricated by embedding an X-shaped resistive strain sensor along the inner surface of the silicone infundibulum [23]. The four sensing branches are oriented orthogonally so that each experiences a distinct deformation trajectory when the suction cup bends, stretches, or tilts during contact. Because negative pressure causes the central dome to collapse asymmetrically depending on the loading direction, each branch generates a characteristic resistance change, enabling the system to resolve local curvature and directional strain. The suction cup is mounted on a load cell to simultaneously measure suction force and strain-sensor output (Figure 3a(ii)). During the preload, attachment, and detachment phases, the sensor signals capture the transition between symmetric compression, stable holding, and rapid elastic recovery, providing multi-dimensional feedback about seal formation and load direction.

The optical-pattern-based sensing module in Figure 3b(i) employs a camera positioned above a patterned internal surface to track deformation-induced pixel shifts in real time [34]. When the suction cup forms a seal, the infundibulum surface deforms non-uniformly, producing measurable changes in marker geometry, brightness, and relative position. This optical configuration effectively converts the continuous deformation field of the suction chamber into a high-resolution spatial map. As illustrated in Figure 3b(ii), pixel-wise RGB difference maps between the undeformed and suctioned states reveal the direction and magnitude of local displacement. Distinct regions of increased or decreased color intensity indicate tilt, off-axis loading, indentation depth, or partial sealing. This imaging-based method captures two-dimensional deformation patterns that cannot be isolated through single-point strain sensing, enabling precise estimation of contact asymmetry and surface curvature.

The design shown in Figure 3c(i) incorporates a fully coated carbon-nanotube (CNT) sensing layer on the outer surface of the suction cup to achieve omnidirectional tactile perception [22]. The CNT-elastomer composite forms a percolated conductive network whose resistance changes sensitively with external strain or shear deformation. Because each region of the cup wall experiences different stress levels when the gripper is pulled sideways, tilted, or pressed eccentrically, the distributed CNT layer outputs spatially distinctive resistance patterns. The simulation in Figure 3c(ii) visualizes how stress concentrates in specific sectors under directional loading, and the associated nine-zone mapping scheme illustrates how the sensor output can be segmented to infer the direction of the applied force. This spatially distributed tactile sensing allows the suction cup to detect misalignment, rim leakage direction, and lateral shear forces—functionalities that are particularly important when interacting with curved or irregular objects.

### 3.2. Arm-Based and Hybrid Sensor-Integrated Modules

Soft robotic arms inspired by octopus tentacles provide manipulation capabilities that complement suction-based gripping by enabling bending, twisting, wrapping, and distributed whole-body contact [36,37]. Unlike suction cups, which primarily rely on normal-direction adhesion and localized deformation, arm-based systems leverage continuum mechanics to conform to a wider range of shapes and interaction scenarios. Prior studies have demonstrated various configurations of octopus-inspired arms—including proprioceptive tentacles with embedded stretchable sensors [24], membrane-based underwater manipulators that couple suction with flow sensing [39], triboelectric-integrated gripping arms capable of frictional state detection [26], and hybrid actuator arrays governed by suction-triggered mechanical logic [17]. These arm and hybrid modules provide sensing modalities—such as curvature estimation, hydrodynamic field detection, triboelectric feedback, and decentralized pneumatic switching—that are not achievable through suction mechanisms alone. By combining distributed actuation with embedded sensing, these systems enable soft grippers to perform adaptive, contact-rich manipulation in environments where surface roughness, geometry, and external perturbations can vary significantly [49]. Arm-based soft grippers extend the functional range of octopus-inspired designs by combining suction with continuum arm deformation, enabling grasping modes that involve bending, wrapping, and distributed contact [17,26,36,37,40,51]. As shown in Figure 4a(i), the tentacle arm developed by Xie et al. [24] integrates four stretchable strain sensors arranged in a crosswise pattern along the arm’s length. Each sensor consists of a silicone-based conductive composite that elongates proportionally with local curvature. Because the arm is pneumatically actuated through internal chambers that inflate asymmetrically depending on the input pressure, different regions of the arm undergo different curvature magnitudes. The crosswise sensor geometry captures both longitudinal stretching and circumferential bending. During actuation, the sensors produce distinct resistance profiles that reflect arm posture. For example, when the arm bends left, the left-side sensor exhibits compressive strain (resistance decrease), whereas the right-side sensor stretches (resistance increase). When the arm performs compound bending with elevation plus lateral curvature each of the four sensors produces a characteristic waveform. In Figure 4a(ii), these resistance patterns are mapped to a curvature reconstruction model, derived from the geometric relationship between sensor elongation and arc angle. This allows continuous estimation of tip pose and arm shape without external cameras. Experiments show that the reconstructed curvature closely matches the measured bending radius (error < 6%), demonstrating reliable proprioception even under asymmetric loading or external disturbances.

The design in Figure 4b(i) is inspired by the glowing sucker octopus *Stauroteuthis syrtensis*, whose umbrella-like membrane enhances adhesion in deep-sea, low-light conditions [40]. Wu et al. reproduced this morphology by combining a flexible radial membrane with a distributed array of micro-suction mouths. A flowmeter mounted upstream monitors the hydrodynamic disturbance generated as the membrane expands or oscillates underwater. When the arm moves, the membrane’s radial deformation compresses and expands surrounding water, generating distinct flow signatures. Approaching a flat surface creates a rapid rise in backflow resistance, while approaching a curved object yields asymmetric flow patterns. In Figure 4b(ii), computational fluid dynamics simulations illustrate how velocity gradients (0.39–11.22 m/s) and pressure distributions (up to ±4 × 10^4^ Pa) evolve around the membrane. These hydrodynamic cues allow the system to infer object proximity, surface shape, and orientation before forming a suction seal, enabling pre-contact sensing in murky or cluttered underwater environments. Because this information is available even when optical sensing fails, the design offers a biologically meaningful alternative for underwater robotic exploration.

As shown in Figure 4c(i), Chen et al. integrate triboelectric tactile sensors into a hybrid arm–suction gripper [26]. The triboelectric layers consist of microstructured elastomer surfaces paired with conductive electrodes, forming a contact–separation mode TENG (triboelectric nanogenerator). When the arm approaches an object, the layered surfaces experience sliding friction and compressive deformation, inducing charge transfer between the materials. The three gripping states shown in Figure 4c(ii) reflect distinct triboelectric outputs: minimal charge signal due to negligible contact (ready state), increased sliding and partial compression generate stronger voltage peaks, revealing frictional engagement and initial contact angle (intermediate state). large-area deformation saturates the signal, corresponding to stable grasp formation (full gripping state). The output waveforms reliably distinguish these transitions, enabling the arm to detect whether the object is fully secured or requires re-engagement. Because triboelectric sensing directly encodes friction state and shear-induced deformation, it provides tactile information unavailable from suction pressure alone. This makes hybrid gripping more robust when handling objects with friction-sensitive surfaces or irregular geometries.

The architecture in Figure 4d(i) uses suction-triggered switches (STSs) distributed across an array of arm segments [51]. Each STS comprises a chamber membrane that deflects when suction occurs, mechanically opening or closing a pneumatic channel. This yields binary signals that propagate to neighboring segments. In Figure 4d(ii), these local pressure-driven signals activate adjacent actuators sequentially: once one suction cup seals onto an object, its STS triggers bending of the next arm segment, which then forms its own seal and activates the next. This cascading logic produces autonomous grasp formation. The multi-step sequence shown in Figure 4d(iii) demonstrates how the arm array encloses the object. In Figure 4d(iv), this distributed embodiment enables full object encapsulation without centralized electronic control. The system relies entirely on mechanical logic encoded in suction events, enabling fast and robust adaptation in irregular environments. This mechanism is analogous to biological reflex pathways, where local sensory events drive immediate motor responses.

Although these suction- and arm-integrated modules provide reliable attachment, conformal deformation, and rich local sensing of mechanical or hydrodynamic cues, their perceptual capabilities remain tightly coupled to the physical interaction itself [49]. They can describe how the structure is deforming, whether a seal is stable, how frictional or flow conditions are changing, or how an arm segment is bending, but they do not yet transform this information into broader interpretive or predictive insight. As a result, they respond effectively to contact but still cannot regulate motion deliberately, refine alignment autonomously, or select grasping strategies based on an understanding of the object or environment. Achieving these functions requires control frameworks capable of integrating sensor feedback into closed-loop decisions, as well as computational approaches that can interpret multimodal signals, classify contact conditions, infer object properties, and guide manipulation at a higher semantic level [27,28,31]. These needs motivate the subsequent section, where sensing is no longer used solely to react to deformation but becomes the basis for coordinated control, object recognition, and data-driven reasoning that elevate soft grippers from mechanically adaptive systems to perceptually and computationally intelligent manipulators.

## 4. Control Frameworks and Computational Intelligence in Soft Grippers

Soft robotic grippers equipped with structural and embedded sensing modules can detect deformation, seal integrity, hydrodynamic cues, and frictional transitions during contact, but these signals alone do not determine how the system should move, adjust its pose, or plan subsequent actions. To translate such raw sensing information into purposeful manipulation, feedback control frameworks are required to regulate motion and refine grasp alignment based on continuous sensor data [1]. Beyond low-level adjustments, higher-level interpretation of tactile, flow-based, and visual cues enables the system to infer contact states, recognize interaction patterns, and distinguish object properties that influence manipulation strategies [28,31]. Recent developments have therefore integrated sensing with machine learning and data-driven inference, allowing soft grippers to move beyond purely reactive responses and toward reasoning about their interactions using multimodal information [27,33]. This section examines approaches that couple feedback control with computational perception to achieve deliberate, adaptive, and context-aware manipulation.

### 4.1. Feedback-Controlled Grasping and Alignment

Soft robotic grippers rely heavily on continuous feedback to ensure stable attachment, compensate for misalignment, and regulate contact forces during interaction. Unlike rigid manipulators that depend on precise kinematics, soft structures deform continuously, making real-time sensing essential for maintaining grasp stability and correcting errors induced by compliance. Embedded sensors such as strain gauges, pressure transducers, triboelectric layers, and flow-based detectors provide measurements that reflect the immediate interaction state, including seal formation, surface tilt, shear-induced deformation, and partial detachment [22,23,26]. By integrating these signals into closed-loop control frameworks, soft grippers can autonomously refine their pose, adjust contact pressure, and reorient themselves to achieve more secure and stable grasping, even in the presence of unpredictable disturbances or irregular surfaces [1]. This section explores representative approaches that utilize multimodal sensor feedback to guide motion correction, stabilize grasp formation, and coordinate manipulation trajectories in real time, forming the control foundation upon which higher-level perception and AI-based inference can operate [49].

Sensor-driven closed-loop control has become an essential mechanism for achieving reliable and adaptable grasping in soft robotic systems. As shown in Figure 5a(i), a multi-chamber suction module equipped with internal pressure sensors is used to detect subtle variations during contact formation and alignment [25]. Each chamber measures pressure fluctuations as the interface approaches, seals, or partially loses contact with the target surface, generating a spatial pressure pattern that reflects leakage direction, surface tilt, and local curvature. These distributed measurements serve as the sensory front end of a feedback loop that continuously evaluates how well the suction cup is aligned with the object. The operation of this loop is captured in Figure 5a(ii). Over time, the four pressure signals evolve as the manipulator cycles through approach, tentative contact, search, and confirmed grasp states. Immediately after initial contact, the controller observes an asymmetric pressure profile: one chamber exhibits a lower vacuum level than the others, indicating a leak on that side. This pressure imbalance is interpreted as an error vector pointing toward the edge of the seal. The control algorithm converts this error into corrective motions which are typically small lateral shifts or in-plane rotations of the suction cup commanded until the pressure differences are reduced below a prescribed threshold. When the seal becomes more uniform, the four pressure traces converge, and the controller switches from a searching state to a holding state, maintaining pump output while monitoring for new deviations. If a subsequent disturbance or object motion reintroduces asymmetry, the same mechanism triggers another micro-adjustment or, in extreme cases, a reseating maneuver. In this way, the suction module executes a closed-loop haptic search: sensor readings define a real-time error signal, the controller maps this error to motion commands, and the resulting pose change is immediately re-evaluated through updated pressure patterns. The system does not simply report whether suction is present; it actively steers the gripper toward a configuration of maximum sealing quality.

A similar sensing–control coupling appears in the tentacle-type arm illustrated in Figure 5b(i) [24]. Crosswise stretchable strain sensors are laminated along the arm to provide a distributed measurement of local curvature and axial stretch. During operation, a desired bending profile such as a target tip angle or curvature distribution is specified, and the controller compares this reference with the posture reconstructed from the strain data. When external contacts, payload changes, or internal pressure drift cause the arm to deviate from the target profile, the error is mapped to corrective pressure commands for the internal chambers. Figure 5b(ii) shows that this process allows the arm to track complex bending trajectories: as the arm moves through a sequence of postures, the reconstructed curvature closely follows the reference, and the feedback loop compensates for compliance-induced overshoot or unmodeled disturbances. In cluttered environments, unexpected collisions create localized strain spikes, which the controller interprets as contact events and responds to by reducing actuation pressure or adjusting the arm’s path to maintain a stable grasp while avoiding excessive contact forces.

The hybrid suction–flow module in Figure 5c(i) employs hydrodynamic sensing as the feedback signal for underwater approach control [40]. A flowmeter is connected to a network of suction discs inspired by the umbrella-like structure of deep-sea octopus suckers. As the gripper moves through water toward an object, the volumetric flow rate and pressure drop across the network change systematically with distance, orientation, and partial occlusion of the suction mouths. These variations provide a continuous measure of proximity and alignment even before contact is established. In Figure 5c(ii), simulations of velocity and pressure fields show that specific flow signatures correspond to safe approach zones, imminent contact, or excessive impact conditions. The controller uses these signatures as feedback: when the measured flow rate falls within a predefined window associated with safe sealing, the approach is slowed and suction is engaged; if the flow indicates rapid occlusion or strong back-pressure, the controller reduces speed or adjusts the trajectory to prevent hard collisions. Once suction is established, deviations in flow can signal partial detachment or leakage, prompting the system to reorient or regrip the object.

These examples demonstrate how soft grippers convert rich, localized sensor data such as pressure distributions, strain fields, and hydrodynamic signals into explicit control variables that drive motion correction and grasp stabilization in real time. Rather than executing open-loop trajectories, the manipulators continuously compare desired interaction states with measurements from embedded sensors, compute error signals, and update actuation commands accordingly. This feedback-control layer provides the necessary foundation for the next level of capability, where data-driven models and AI-driven decision-making become essential to coordinate complex actions, interpret multimodal sensing, and achieve more deliberate and predictive control behaviors.

### 4.2. Data-Driven Perception and AI-Based Contact-State Recognition

While feedback control enables soft grippers to correct alignment and stabilize contact in real time, achieving more autonomous and context-aware manipulation requires the ability to interpret sensor signals rather than merely react to them [28]. Soft grippers generate complex, high-dimensional data streams from embedded strain sensors, pressure arrays, optical markers, and hydrodynamic detectors [31,41]. These signals encode object-dependent properties such as stiffness, curvature, material type, and contact transitions, but the relationships are nonlinear and often too intricate to analyze using rule-based or model-driven control alone [1]. AI- and machine-learning-based approaches have therefore become essential for extracting meaningful patterns from these deformation and pressure signatures [27]. Recent studies demonstrate that learned models can classify contact states, detect subtle interaction phases, and infer object characteristics directly from sensor data, achieving levels of perception that exceed hand-designed heuristics [33]. By transforming raw tactile and deformation signals into semantically meaningful representations, AI-driven perception modules form a critical layer between low-level sensing and higher-level manipulation planning, enabling soft grippers to recognize what they are touching and adapt their strategies accordingly.

Machine-learning-based perception provides the intermediate layer between low-level sensor feedback and the high-level AI inference described later. In contrast to simple threshold-based control, these approaches learn deformation–signal relationships from large datasets and generalize them across objects, materials, and interaction conditions. Data-driven perception enables soft grippers to convert complex deformation and pressure signals into interpretable contact states. As shown in Figure 6a(i), the deformation-driven recognition module collects distributed strain or pressure data from the suction cup or soft arm during grasping [22]. When interacting with an object, the sensor array records a time series of mechanical responses that reflect not only whether contact has occurred, but also how the structure is bending, how the seal is forming, and how the load is distributed across the interface. These sensor patterns differ depending on the geometry, stiffness, and surface properties of the contacted object. For example, stiff objects induce sharper and more localized strain peaks, while compliant objects produce lower-magnitude but more diffuse signatures. Surface curvature also influences the spatial distribution of pressure gradients, generating distinct mechanical profiles detectable through the sensing layer. The interpretation of these patterns is handled by a machine-learning classifier, illustrated in Figure 6a(ii). Raw time-series sensor data are first transformed into feature vectors—such as peak amplitude, spatial gradients, temporal derivatives, or frequency-domain descriptors—which form the input to supervised learning models. These models are trained to discriminate among predefined contact conditions, including object categories, surface orientation, and stable versus unstable grasp states. During inference, the classifier maps the current sensor signature to the most probable contact state. Because the mapping between deformation patterns and physical states is highly nonlinear and coupled across multiple sensing channels, AI models outperform rule-based logic and provide robust generalization even under environmental noise or partial occlusion. The closed-loop control sequence shown in Figure 6a(iii) demonstrates how the robot actively refines its positioning using continuous sensory feedback. After approaching the surface and establishing initial contact, the system performs a primary alignment step to correct coarse positional errors based on the first set of sensor cues. This is followed by a secondary, finer alignment phase that adjusts the gripper pose with higher spatial precision. Once these two alignment stages converge, the robot initiates a center-search routine that incrementally probes the surrounding region to locate the optimal gripping point. Through this sequential process, the robot progressively reduces positional uncertainty and ensures that the suction interface is properly centered before actuation. This multistage feedback loop highlights how structural sensing and closed-loop computation cooperate to achieve reliable grasping performance in unstructured settings.

A complementary perception framework is shown in Figure 6b(i), where strain-derived waveforms are used not for alignment but for material stiffness recognition [23]. During indentation, different materials—ranging from Soft to Hard—produce distinct temporal waveforms because softer objects result in longer-duration, lower-amplitude deformation, whereas harder objects induce sharper peaks and more abrupt transitions. These waveform segments are processed by a classifier that assigns each trial to one of the four stiffness categories. Beyond material classification, the same distributed sensor signals are used to estimate the contact angle. As shown in Figure 6b(ii), the regression model predicts the inclination angle with high fidelity by learning the nonlinear mapping between strain signal morphology and geometric orientation. Figure 6b(iii) combines stiffness classification with angle estimation in a matrix form that demonstrates the ability to simultaneously determine both an object’s compliance class and its contact geometry with accuracy values between 94% and 100%.

The highest perceptual resolution is achieved using optical tactile imaging, as shown in Figure 6c(i). A ChromaTouch marker layer inside the suction chamber undergoes deformation during the Neutral, Inflation, Indentation, Suction, and Pick-up phases [34]. A camera captures these deformation fields, producing structured tactile images whose pixel-level distortions reflect contact position, object curvature, and sealing quality. The pressure–time curve plotted alongside the images provides synchronized internal pressure feedback useful for detecting suction formation and detachment events. A CNN processes the marker-deformation images and infers the inclination angles (θ, φ) of the object or surface. Finally, Figure 6c(ii) shows how these predictions are embedded into a closed-loop manipulation cycle: the gripper first indents the surface, obtains the predicted angles, autonomously corrects its orientation, forms a complete seal, and executes a stable pick-up.

AI-driven tactile inference enables soft grippers to interpret rich electromechanical signals and extract semantic information about object properties and interaction dynamics. During grasping, the triboelectric membrane generates voltage signals through both normal indentation and subtle lateral micro-sliding, producing time-domain patterns that evolve across the sealing–enveloping–release cycle. These waveform transitions, which can be observed in the example sequences of Figure 7a(i), encode geometric and surface-dependent characteristics of the contacted object [26]. When signals from the four distributed sensing units are visualized as 3D trajectories, distinct amplitude–phase–frequency combinations emerge for each object class, even under identical loading conditions (see Figure 7a(ii)). A convolutional neural network extracts these temporal–spatial features automatically—without the need for manual feature engineering—and uses them to perform multiple recognition tasks. The model accurately distinguishes stiffness levels among Ecoflex, Dragon Skin PX, and Dragon Skin 30 with an accuracy of 98.0% (Figure 7a(iii)), leveraging the triboelectric principle that stiffer materials yield stronger charge separation. A similar approach successfully classifies spheres, cubes, cones, and vertical/horizontal cylinders with 98.3% accuracy (Figure 7a(iv)). Beyond synthetic objects, triboelectric envelopes generated by biological models such as sea cucumbers, starfish, and crabs also show separable patterns, enabling generalization to irregular natural geometries (Figure 7a(v)).

A complementary form of intelligence arises from the hierarchical suction architecture. Instead of relying solely on pressure values, the system integrates mechanical events triggered by suction-triggered switches with high-level AI interpretation. As illustrated in Figure 7b(i), STS units provide rapid binary indicators of sealing or detachment, initiating transitions between extension, suction engagement, and retraction [51]. On top of this physical switching layer, a computer–pressure sensor module analyzes continuous pressure trends to infer environmental conditions. The grasp sequences under varying surface states, including dry/wet conditions and different grit roughness levels, demonstrate how these multi-tier signals evolve in real time (Figure 7b(ii)). The resulting pressure–time curves, shown in Figure 7b(iii), exhibit sharp transitions at initial contact and condition-dependent decay profiles linked to surface roughness, wetness, and applied load. These single-channel pressure signatures contain surprisingly high information density. When processed by an AI classifier, they enable reliable recognition of material conditions, as confirmed in the confusion matrix of Figure 7b(iv). The system’s ability to infer multiple environmental properties from a single pressure signal exemplifies the efficiency of suction-mediated sensing and highlights the emergent behavior that results from combining mechanical switching with data-driven interpretation.

These AI-based modules transform raw triboelectric and pressure signals into high-level semantic understanding [27]. The gripper can recognize object identity, classify stiffness and shape, detect environmental characteristics, and even anticipate failure-prone configurations. Such perceptual abstraction moves soft grippers beyond reactive contact detection, enabling predictive and context-aware manipulation that forms the upper tier of intelligent soft robotic behavior [49].

## 5. Challenges and Future Directions for Intelligent Soft Grippers

Although octopus-inspired grippers have advanced from structural replication to perception-driven manipulation, significant challenges remain before these systems can achieve the level of adaptive, real-time intelligence required for practical deployment [1]. Early suction cups and soft arms successfully demonstrated reliable adhesion and basic proprioception through geometric and material mimicry [49], but their capabilities were fundamentally reactive and limited to controlled conditions [55,56,57,58]. Subsequent perception-integrated designs introduced multimodal sensing and machine learning, enabling recognition of material stiffness, surface orientation, and contact geometry through pressure signals, strain responses, and optical deformation fields [22,23,28,54,59]. These developments marked an important shift toward context-aware manipulation yet revealed persistent limitations in environmental robustness, energy efficiency, and sustained performance outside the laboratory [39,60,61,62,63]. Even with advanced sensing and AI-enhanced perception, many systems still struggle with generalization across diverse materials, resistance to contamination, and reliable operation under wet, abrasive, or variable lighting conditions [39].

A central barrier lies in the integration of sensing, computation, and control within deformable bodies. Embedding multimodal sensors into elastomers creates mismatches in stretchability, modulus, and adhesion that lead to signal drift, delamination, micro-leakage, or mechanical interference over repeated loading cycles [31,54,64,65]. High-density sensor arrays also introduce challenges in wiring, packaging, and maintaining consistent contact with soft substrates during dynamic deformation [41,59,66,67,68,69]. These issues are amplified at system scale, where hybrid structures must coordinate soft membranes, fluidic actuation, rigid connectors, and distributed electronics without compromising compliance or structural integrity [1,3,54,70,71]. Fatigue in cyclic suction can degrade seal performance, while conductive composites such as CNT networks or metal traces may fracture under torsion or repeated shear loading [23,59,72,73]. Durability and long-term reliability therefore remain key barriers to real-world applications.

Computation presents another constraint. High-bandwidth tactile imaging, multi-channel strain acquisition, and deep-learning inference generate significant latency and energy consumption, which limits closed-loop responsiveness in fast manipulation tasks [22,61,74,75]. The difficulty of synchronizing high-frequency signals from heterogeneous sensors further contributes to processing bottlenecks [31,75]. Many systems depend on external computation resources or cloud-based inference, which restricts portability and precludes deployment in time-critical or resource-constrained environments [27]. Power-hungry pumps, onboard illumination for optical systems, and continuous neural-network inference additionally raise overall energy demands, underscoring the need for lightweight, energy-efficient processing approaches [22].

These challenges motivate future directions that require reconsidering how intelligence should be embedded within soft robotic architectures. Approaches inspired by neuromorphic principles including event-driven sensing, parallel signal pathways, and localized computation offer a path toward low-power, real-time operation, yet remain technically immature [51,73,75]. Unlike traditional perception pipelines relying on dense sampling and centralized processing, neuromorphic-inspired architectures emphasize distributed intelligence where sensing and computation occur in close physical proximity. Fluidic logic circuits and snap-based mechanisms have already demonstrated rapid, decentralized decision-making for suction initiation and curling upon contact [39], while optoelectronic tactile modules and triboelectric systems integrated with deep learning illustrate how local signal encoding can support adaptive behavior [26,33,51,65]. However, these efforts remain proof-of-concept and require advances in flexible neuromorphic hardware, high-density event-based sensors, and fabrication processes that allow integration of processors and sensing elements on a unified substrate [27,76].

Future systems should combine multimodal sensing arrays with specialized AI hardware to enable efficient edge processing, reducing latency and power consumption while supporting real-time contextual reasoning [27,75,76]. Energy-harvesting materials such as triboelectric nanogenerators and piezoelectric harvesters may provide self-powered operation in portable or long-duration missions [26,62,63,76]. Intelligent suction strategies capable of dynamically adjusting adhesion in response to roughness, wetness, contamination, or material variability are essential for operation in unstructured environments [3,39,51,56,63,71]. Addressing these capabilities will also require advancing packaging strategies that improve washability, chemical resistance, skin compatibility, and long-term stability, as well as balancing trade-offs among sensitivity, stretchability, hysteresis, linearity, and power consumption for specific use cases [31].

Beyond technical refinement, practical validation remains a pressing gap. Many promising systems have not been evaluated outside controlled laboratory conditions, limiting understanding of their true robustness and failure modes [39,55,74,77,78]. Future testing protocols should include long-term cyclic durability, adhesion performance under varying temperature and humidity, resistance to dust and contamination, and operation under complex mechanical disturbances [3,23,75]. Demonstrating value in real-world scenarios—such as disaster response, medical robotics, and deep-sea or space exploration—will be critical to establishing both technological and societal relevance [3,39,49,57,79]. Achieving the full potential of intelligent octopus-inspired grippers will require interdisciplinary framework across materials science, soft robotics, neuroscience, and AI hardware design, with system-level convergence of sensing, computation, and control as a central objective.

## Figures and Tables

**Figure 1 biomimetics-10-00813-f001:**
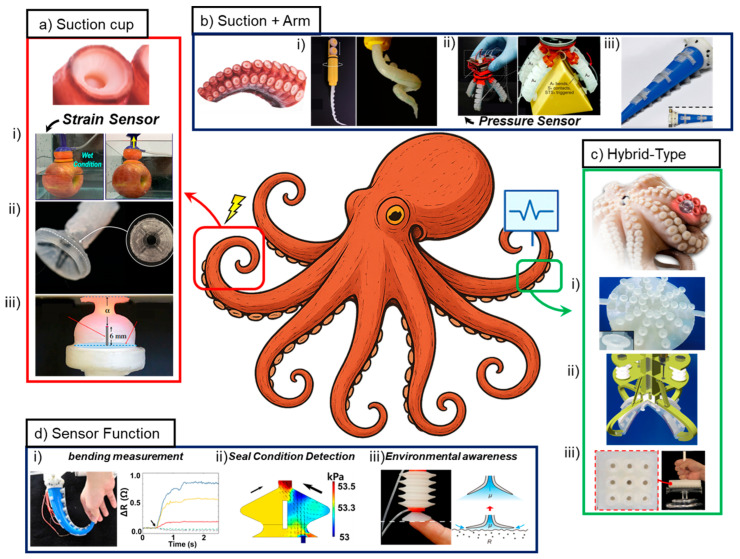
Comprehensive overview of octopus-inspired grippers and tactile sensing systems, illustrating the evolution from suction-based adhesion to multi-arm manipulation and integrated sensing. (**a**) Grippers mimicking a single suction cup structure (adapted from [21,22,23]). (**b**) Grippers inspired not only by suction cups but also by the octopus arm morphology, enabling enhanced grasping through soft tentacle designs (adapted from [17,24,51]). (**c**) Multi-arm octopus-inspired grippers equipped with arrays of suction structures (adapted from [14,26,40,52]). (**d**) Octopus-inspired grippers integrated with tactile sensors, demonstrating the role of sensing (adapted from [24,25,51]). (i) External stimuli cause section-dependent resistance changes: significant increases in elongation sensors versus slight decreases in expansion sensors. (ii) A vertical leak induces high-velocity airflow within the specific chamber, creating a distinct pressure difference that allows for the localization of the leak. (iii) The suction gripper on a rough surface draws in surrounding fluid in the direction of the arrow to evaluate the degree of roughness.

**Figure 2 biomimetics-10-00813-f002:**
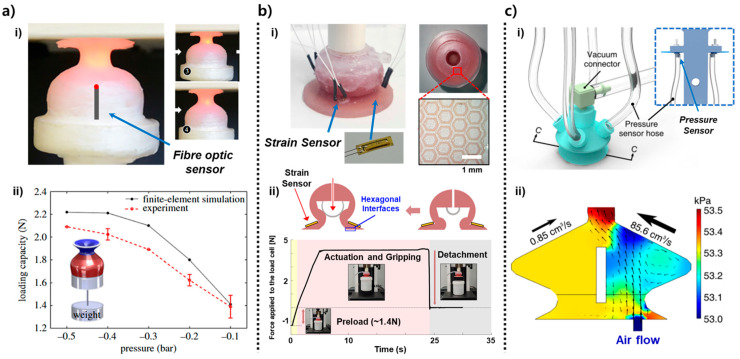
Structural principles and representative implementations of suction-based soft grippers. (**a**) Anchoring-type suction cup with stiffness-gradient infundibulum and optical sensing of deformation (adapted from [21]). (i) The infundibulum is fabricated with a stiffness-gradient structure, and an optical fiber-based proximity/tactile sensor is embedded inside the sucker. The sensor measures anchor length during infundibulum deformation in real time, recording 6 mm and 8 mm for cases 3 and 4, respectively (ii) A graph of holding force versus vacuum pressure (–0.1 to –0.5 bar) between experimental data and finite element analysis (FEA). As the negative pressure increases, the attachment holding force increases, and the sensor detects the elongation of anchor length in response to increased load. (**b**) Hybrid friction–suction structure with patterned interfaces and strain-sensor-based deformation monitoring (adapted from [54]). (i) Structure of a hierarchical soft gripper where hexagonal friction patterns and strain sensors are arranged around the contact interface. The patterns enhance shear resistance, while the sensors detect asymmetric deformations in adjacent directions to monitor gripper–object interactions. (ii) Force–time curve measured during preload, suction, holding, and detachment sequence. After an initial preload of approximately 1.4 N, stable attachment force is maintained, followed by a sharp force drop during detachment. Deformation at each stage is confirmed through variations in sensor signals. (**c**) Multi-chamber smart suction cup utilizing internal pressure differentials for shape/orientation recognition (adapted from [25]). (i) 3D model of a smart suction cup with multi-chamber configuration and embedded internal pressure sensors. The pressure sensors measure micro pressure distribution changes between chambers in real time to estimate object geometry and contact conditions. (ii) Simulation results showing chamber internal pressure distribution (kPa) under tangential air flow. By analyzing flow-rate-based variations, the seal leak direction can be estimated, enabling edge detection of objects in haptic object search algorithms.

**Figure 3 biomimetics-10-00813-f003:**
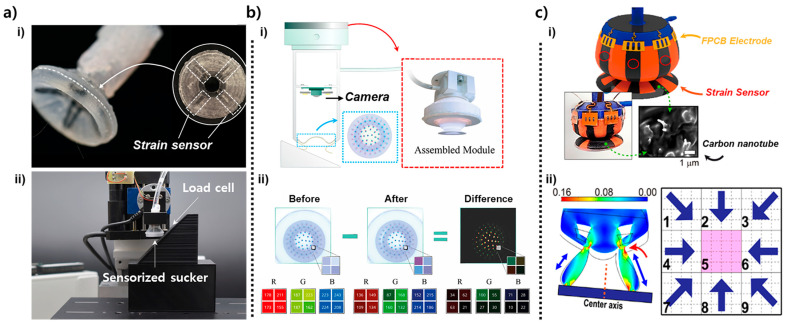
Soft suction modules integrating deformation sensing, optical monitoring, and CNT-based tactile perception. (**a**) Strain-sensorized suction cup for detecting loading direction and adhesion state (adapted from [23]). (i) Structure of a silicone-based artificial suction cup embedded with a cross (X)-shaped strain sensor, which converts localized deformation around the infundibulum into electrical signals to detect contact state and loading directionality. (ii) Pull-off experimental setup using a load cell and vertical indentation stage, measuring the force variations occurring during attachment, compression, and detachment of the suction cup. (**b**) Optical deformation sensing using camera-based internal marker tracking for high-resolution contact mapping (adapted from [34]). (i) Configuration integrating a camera and an LED ring on the upper side of the suction cup. The displacement of internal color/marker patterns is captured to acquire high-resolution RGB images of subtle contact-induced deformation. (ii) Reconstruction of deformation fields through RGB difference analysis of pre- and post-suction images. During suction, pattern displacement vectors change, enabling calculation of contact depth, directionality, surface inclination, and fine offset deformation. This approach has lower drift and higher resolution compared to conventional strain-sensor methods, making it suitable for object shape estimation and precise pose recognition. (**c**) CNT-coated suction cup for multi-directional strain perception and distributed contact evaluation (adapted from [22]). (i) A 3D structure combining multiple attachment pads, FPCB electrodes, and piezoresistive CNT composite coating applied to the outer wall. The CNT network detects micro shear and bending deformation in real time, allowing estimation of eccentric loading, center-axis deviation, and subtle surface inclination. (ii) Finite element analysis (FEA) results of stress concentration regions during suction and the 9-zone contact mapping model. The FEA contour plot uses color to represent the strain percentage (Strain [%]) concentrated on the infundibulum. The 9-zone model uses numbers (1–9) to denote contact regions and arrows to indicate the predicted feedback vector toward the object’s Center of Gravity (CG). Using strain sensors positioned at the outer rim of the suction cup, the system infers the direction of applied forces and determines whether contact is centered or shifted toward the periphery.

**Figure 4 biomimetics-10-00813-f004:**
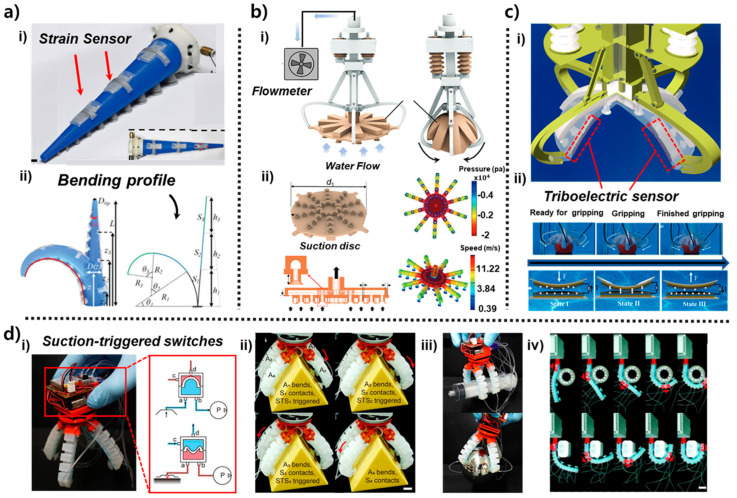
Arm-type, hybrid, and wide-area suction grippers with embedded sensing for adaptive grasping. (**a**) Proprioceptive tentacle actuator distinguishing active vs. passive bending (adapted from [21]). (i) Tapered soft tentacle with an attached EL–EP cross-type stretch sensor. By separately measuring elongation and perimeter expansion under inflation, it distinguishes actuator-driven bending from external load-induced bending in real-time. (ii) Experimental comparison confirms that curvature and suction states are accurately reconstructed using only sensor signals during free bending, load bending, and suction grasping. (**b**) Wide-area suction disk for underwater grasping and pressure sensing (adapted from [36]). (i) Umbrella-shaped soft disk with a funnel-shaped mouth array, enabling stable attachment on rough surfaces via distinct pressure distributions during contraction/expansion. (ii) Comparison of experimental and Finite Element Method (FEM) analysis for suction force–pressure distribution. Shifting suction from center to periphery reveals higher stability and flow variation in central regions. (**c**) Triboelectric tactile sensing in suction modules for underwater recognition (adapted from [23]). (i) Soft gripper integrating triboelectric nanogenerators (TENG) within the suction cup network. MXene-based materials ensure high sensitivity for detecting pressure, separation, shape, and hardness underwater. (ii) Real-time grasping experiments achieved classification accuracies of 98.3% (shape) and 98.0% (hardness), verifying the capability to monitor grasping states via sensor patterns. (**d**) Distributed fluidic circuitry enabling embodied intelligence (adapted from [47]). (i) Octopus-inspired hierarchical system combining suction cups, fluidics, and sensors. A suction-triggered switch (STS) autonomously executes suction, curving, and encapsulation by utilizing pressure variations upon contact. (ii–iv) Results for contact detection, roughness classification, medium transition, and pulling force prediction based solely on chamber pressure. This proves embodied multimodal perception is achievable without high-level computation. (Scale bars: (ii) 10 mm, (iv) 20 mm).

**Figure 5 biomimetics-10-00813-f005:**
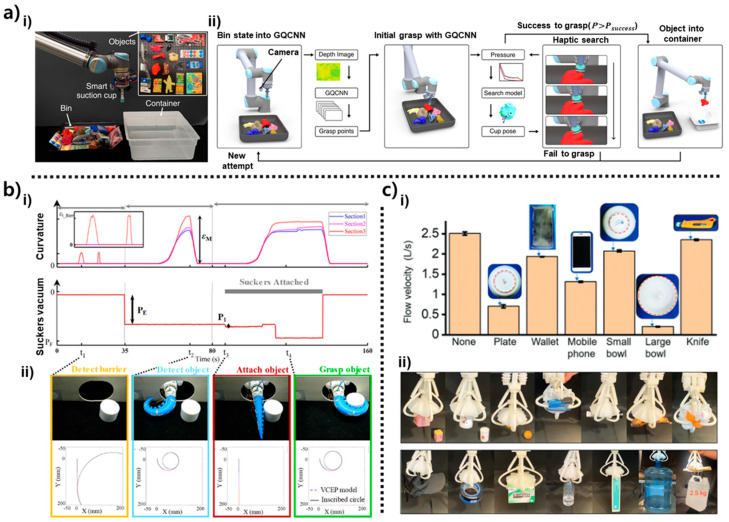
Sensor-guided operational behaviors for feedback control in octopus-inspired soft grippers. (**a**) Haptic-search suction cup combining vision prediction with pressure-based realignment for robust grasping (adapted from [25]). (i) Initial grasping scene where a smart suction cup mounted on a robotic arm interacts with various everyday objects in a cluttered environment. (ii) The bin state captured by the camera is input to Grasp Quality Convolutional Neural Network (GQCNN) to predict an initial grasp pose. (**b**) Tentacle gripper integrated with tactile sensors, demonstrating state detection during obstacle contact, attachment, and grasping (adapted from [24]). (i) Process in which a tentacle structure embedded with a cross-shaped stretchable strain sensor detects curvature (top), along with stepwise sensor responses during obstacle detection, object contact, attachment, and grasping, accompanied by variations in suction chamber pressures ($P_{1}$, $P_{2}$). Section 1 corresponds to the region near the base, and Section 3 corresponds to the region near the tip, of the robotic arm’s three distinct segments. Each sensor measures the local bending of its corresponding section in real time, reducing positional error and enabling stable gripping. (ii) Based on sensing data, the tentacle distinguishes between obstacle (yellow), planar object (blue), attachment (red), and grasping (green) states, with XY-tracking curves visualizing tip position changes at each stage. (**c**) Flow-based tactile sensing for classifying object size and enabling stable grasping in underwater environments (adapted from [40]). (i) Experimental results demonstrating classification of contact object types and sizes by measuring flow rate variations using a turbine flow sensor. (ii) the gripper successfully grasps flat objects, irregular objects, multiple small objects, and heavy objects (up to 2.5 kg) with stability.

**Figure 6 biomimetics-10-00813-f006:**
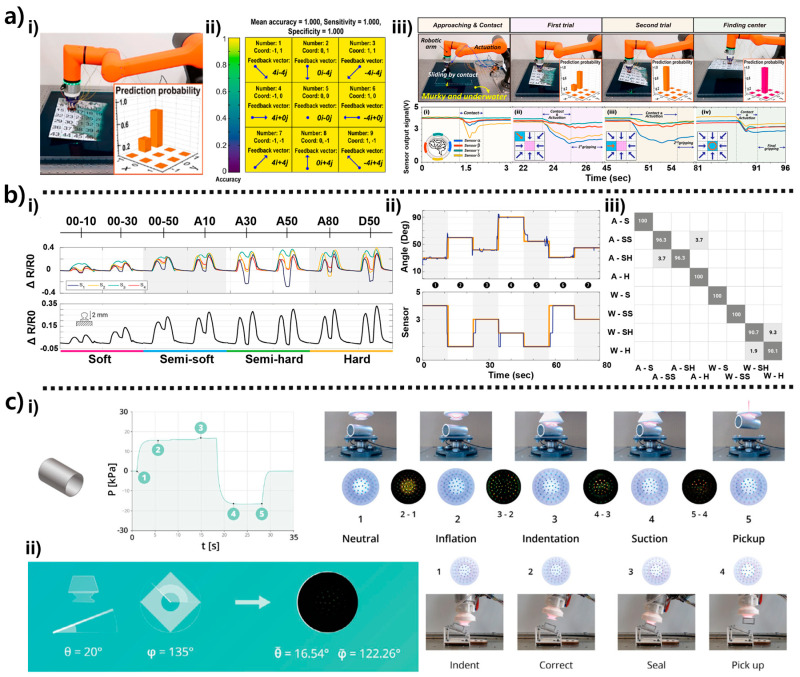
Machine-learning-based perception for object classification, pose estimation, and material recognition. (**a**) Electrical/pressure-based alignment and contact classification using supervised learning (adapted from [19]). (i) Planar alignment experiment where a smart suction gripper measures surface tilt in real-time via strain sensors on the pad’s exterior. (ii) Confusion matrix showing perfect sensitivity and specificity (1.000) confirms the trained AI model robustly distinguishes contact patterns under varying conditions. (iii) The robot executes closed-loop control: approach/contact, primary alignment, secondary alignment, and center search. (**b**) Strain-signal-based stiffness classification and angle estimation via regression (adapted from [20]). (i) Classification of ‘Soft’ to ‘Hard’ materials using ΔR/R_0_ responses from four strain sensors distributed on the cup’s outer wall. (ii) Time-series comparison of sensor resistance (ΔR/R_0_) and ML-predicted contact angles demonstrates that actual angles (orange line) are accurately estimated by the predicted values (blue line) from concurrent signal patterns. (iii) Classification matrix for various material/angle combinations shows accuracies ranging from 94% to 100%. (**c**) Optical tactile imaging and convolutional neural network (CNN)-based pose estimation for autonomous correction (adapted from [30]). (i) Visualization of suction states (Neutral to Pick-up) via camera-captured ChromaTouch marker deformation. Pressure–time curves reflect internal pressure changes, while marker patterns vary with object shape (flat/curved). (ii) Closed-loop sequence (Indentation, Correction, Seal, Pick-up). A CNN predicts inclination (θ, φ) from images, enabling the robot to automatically adjust its pose for stable sealing and reliable pick-up.

**Figure 7 biomimetics-10-00813-f007:**
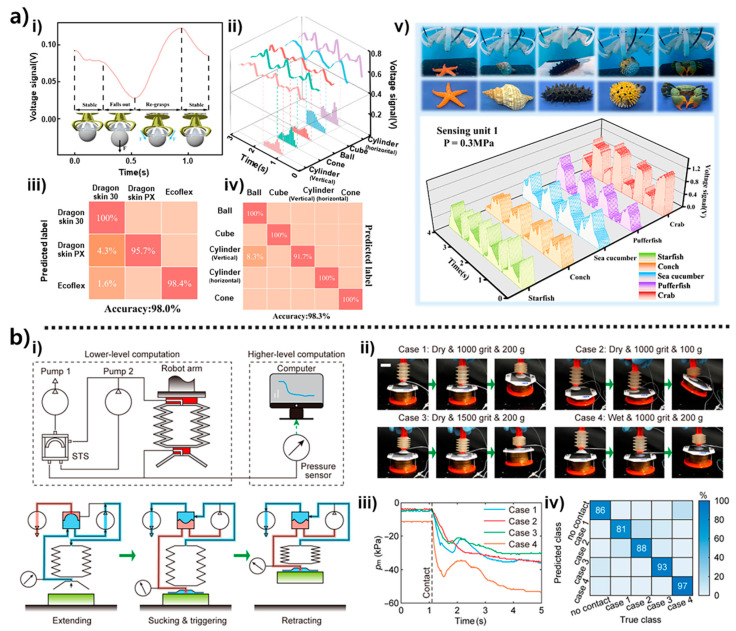
AI-enhanced tactile perception and hierarchical suction intelligence. (**a**) Triboelectric sensing and deep-learning classification for shape, hardness, and biological recognition (adapted from [23]). (i) Time-domain waveforms during grasping of varied shapes (sphere, cube, cone, cylinder) show signal feature changes across the suction–enveloping–release process. (ii) 3D waveforms from four sensing units demonstrate clear separation of voltage patterns for different shapes under identical force, serving as deep learning inputs. (iii) CNN-based hardness classification for three materials (Ecoflex, Dragon Skin PX, Dragon Skin 30) achieved 98.0% accuracy, reflecting triboelectric characteristics where charge generation increases with stiffness. (iv) CNN-based shape recognition (sphere, cube, cone, cylinders) achieved 98.3% accuracy by utilizing relative pattern differences among the four channels. (v) 3D voltage profiles during grasping of marine models (sea urchin, starfish, sea cucumber, clam, crab) confirm that external geometry and surface characteristics are reflected in waveform outputs. (**b**) Suction-triggered switches and multi-level pressure analysis for autonomous action (adapted from [47]). (i) Hierarchical control structure comprising a lower-level suction-triggered switch (STS) and an upper-level computer–pressure sensor module. (ii) Sequential motion images of the suction process under various conditions (dry/wet, roughness, holding loads). (iii) Pressure ($P_{m}$)–time curves show sharp changes at contact, with distinct decay/recovery patterns based on roughness, weight, and wetness serving as AI classifier inputs. (iv) Classification results for contact, roughness, and dry/wet conditions demonstrate that multiple properties are recognized using a single pressure signal, quantitatively proving the core capability of hierarchical suction intelligence.

## Data Availability

Data will be made available on request.

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
