# Peer review of "Design and Sensing Frameworks of Soft Octopus-Inspired Grippers Toward Artificial Intelligence"

_biomimetics, 2025, doi:10.3390/biomimetics10120813_

Round 1
Reviewer 1 Report
Comments and Suggestions for Authors
This article is a review devoted to the evolution of robots that use the principles of grasping parts, as in the tentacles of an octopus.
The article is based on 62 publications, although only about six were cited for the current year, and about a dozen for 2024. The majority of publications are from an earlier period. It follows that either interest in this area has waned or an insufficient number of sources have been analyzed.
It's a bit illogical that the article contains the word "Progress," but its presentation deviates from the chronological order of advances in this field. If the article began with a discussion of early research in this field and then progressed to more and more recent work, then the bibliography would presumably follow some historical chronological order, citing the earliest publications on the topic first, followed by increasingly new publications, and finally, publications from the last two years. This isn't the case.
Unfortunately, the article doesn't live up to its claims. The authors have used excerpts of images and information from other publications, and therefore the illustrations contain some specific information that is irrelevant to the material being presented and, perhaps, unclear even to the authors of this review. For example, Figure 2 contains a wealth of information unrelated to the structure of the suction cups. The same can be said for Figures 3, 4, and 5. The signal graphs, which are not described in the article, only complicate the article's comprehension. If the article were discussing how signals are processed in such devices, then signals in different systems would need to be compared. If the article were discussing how the suction cups are designed, then a single figure should demonstrate various suction cup solutions and analyze which are more promising and why. If the article is discussing how the tentacle structure is implemented—that is, the supporting structure on which the suction cups are located—then these technical solutions should be compared. And if we are talking about how to control these tentacles and the suction cups on them, then we need to devote a chapter to these problems.
Then it would be clear why the bibliography is not in chronological order. For example, one chapter would be devoted to the implementation of suction cups, another to the implementation of tentacles, a third to the implementation of sensors on the suction cups, a fourth to the implementation of sensors for tentacle control, and a fifth chapter could be devoted to the implementation of low-level control, that is, the suction and release of the suction cups upon command from above, the flexion and extension of the tentacle, and its movement upon command from above, and a sixth chapter should be devoted to the implementation of high-level control. If the authors identify any additional problems, a separate section could also be devoted to them. If some problems remain unresolved and their solutions are not covered in the literature, then these deserve a separate discussion. In this case, it would be appropriate to indicate in the final conclusions that much remains to be solved in this area, and it would also be necessary to indicate which of the identified problems have already been sufficiently resolved and therefore do not require further in-depth research.
A reader who has taken the time to read this article deserves to be clearly informed about which problems have been solved, which articles describe the best solutions, which problems have been solved but not effectively, and which problems remain unsolved and require urgent attention. If there are examples of studies that have addressed all of these problems, this deserves a separate mention. Instead, the reader is met with the usual generalities of bad reviews, such as how the topic of this article is highly relevant, how it is rapidly developing, how much remains to be done, and how much is currently being done. These are all unrelated and uninteresting discussions. The article doesn't feel like the authors are truly engaged in the topic of this group of studies; they exist merely as librarians who have collected some articles and compiled a summary from them. This is the level of a student's term paper, not scientific research.
Author Response
We deeply appreciate the reviewer for their thorough reading of our manuscript and useful comments. Our item-by-item response follows:
COMMENTS TO AUTHOR:
Reviewer #1
- This article is a review devoted to the evolution of robots that use the principles of grasping parts, as in the tentacles of an octopus. The article is based on 62 publications, although only about six were cited for the current year, and about a dozen for 2024. The majority of publications are from an earlier period. It follows that either interest in this area has waned or an insufficient number of sources have been analyzed.
â–¶ Thank you very much for this insightful comment. We agree that including recent publications is essential for accurately reflecting the current state of the field. In the initial submission, several key recent studies may not have been sufficiently highlighted due to the emphasis on foundational biological and structural works.
To address this concern, we have taken the following actions.
We have conducted an additional comprehensive literature search and incorporated 22 new publications from 2023–2025, including recent works on:
- AI-integrated suction cups and optical tactile systems (Advanced Intelligent Systems 2024–2025, Sci. Robotics 2025)
- Underwater triboelectric tactile grippers (Nano Energy 2025)
- Hierarchical suction intelligence systems (Science Robotics 2025)
- Classification and evaluation of octopus-inspired suction cups (Advanced Science 2024)
- Event-driven tactile and neuromorphic sensing (Neuromorphic Computing & Engineering 2025)
These newly added references are cited throughout Sections 2–4 of the revised manuscript (e.g., Refs. [34], [37–40], [42], [47], etc., in the updated version).
The field of octopus-inspired grippers draws heavily from foundational biological and mechanical studies (2010–2018), which establish suction mechanisms, muscular hydrostat principles, and soft-material design rules. These core studies remain relevant and are necessary to contextualize the progression toward recently proposed perception-integrated and neuromorphic-oriented systems. We have now explicitly stated this rationale in the Introduction and early sections of the revised manuscript to help readers understand the balance between foundational and contemporary literature.
The revised manuscript now highlights that interest in this field has not waned, but instead has shifted toward: multimodal sensing, AI-driven perception, neuromorphic tactile interfaces, underwater functional grippers, embodied intelligence via soft-fluidic logic. These trends are consistently reflected in the updated subsections (Sections 3 and 4) and through new figures and tables summarizing 2023–2025 works.
We have expanded tables and figure captions to include performance metrics and sensing approaches from the most recent literature. (Example: Table 2 updates sensing strategies and ML performance using 2023–2025 publications.)
We sincerely appreciate the reviewer’s suggestion, which significantly strengthened the manuscript. The revised version now incorporates a broader and more up-to-date coverage of recent research trends and provides clearer justification for the inclusion of earlier foundational works.
- It's a bit illogical that the article contains the word "Progress," but its presentation deviates from the chronological order of advances in this field. If the article began with a discussion of early research in this field and then progressed to more and more recent work, then the bibliography would presumably follow some historical chronological order, citing the earliest publications on the topic first, followed by increasingly new publications, and finally, publications from the last two years. This isn't the case.
â–¶ We appreciate the reviewer’s observation regarding the potential inconsistency between the previous title containing the term “Progress” and the manuscript’s non-chronological structure. To avoid any misleading implication of a historical sequence, we have revised the title to “Design and Sensing Frameworks of Soft Octopus-Inspired Grippers toward Artificial Intelligence.” This revised title more accurately reflects the organizing principle of the review, which is based on functional and conceptual integration rather than temporal progression.
The field of octopus-inspired soft grippers has advanced through multiple parallel and overlapping research streams—for example, suction geometry established in earlier biological studies is continually reused and enhanced with recent sensing technologies, and AI-driven perception methods have emerged independently of structural developments. Because these advances do not follow a strictly linear historical trajectory, a chronological presentation would obscure the technical relationships among structural, sensing, and computational modules. For this reason, the manuscript adopts a framework-based structure that groups studies according to their functional roles—suction mechanisms, sensor-integrated designs, continuum arm systems, and AI-assisted control—allowing readers to understand how these components collectively support the development of intelligent soft grippers.
To ensure that this intention is clear to the reader, we have added an explanatory statement in the Introduction and early Section 3 noting that the manuscript synthesizes research according to functional categories rather than historical phases. We believe this revision, together with the updated title, now resolves the concern raised by the reviewer and provides a coherent rationale for the organizational structure of the review.
- Unfortunately, the article doesn't live up to its claims. The authors have used excerpts of images and information from other publications, and therefore the illustrations contain some specific information that is irrelevant to the material being presented and, perhaps, unclear even to the authors of this review. For example, Figure 2 contains a wealth of information unrelated to the structure of the suction cups. The same can be said for Figures 3, 4, and 5. The signal graphs, which are not described in the article, only complicate the article's comprehension. If the article were discussing how signals are processed in such devices, then signals in different systems would need to be compared. If the article were discussing how the suction cups are designed, then a single figure should demonstrate various suction cup solutions and analyze which are more promising and why. If the article is discussing how the tentacle structure is implemented—that is, the supporting structure on which the suction cups are located—then these technical solutions should be compared. And if we are talking about how to control these tentacles and the suction cups on them, then we need to devote a chapter to these problems.
â–¶ We appreciate the reviewer’s detailed feedback regarding the figures and the relevance of the information included. We fully agree that several visual elements in the initial submission that are particularly signal plots, sub-graphs, and overly detailed excerpts from original publications were not essential to the conceptual narrative of our review and could hinder clarity. In response, we have undertaken a substantial restructuring of the figures and the corresponding text.
In the revised manuscript, all illustrations have been reorganized to align more clearly with the new structure of Sections 3 and 4. Section 3 now focuses solely on the structural and sensing modules of octopus-inspired grippers, while Section 4 presents the control and AI-related operational aspects. Figures previously containing heterogeneous information have been separated, simplified, or completely redesigned so that each figure now supports a single conceptual purpose. Unnecessary signal graphs, force curves, or system-specific details that were not explicitly analyzed in the manuscript have been removed. As a result, Figures 2–5 now provide only the essential structural, sensing, or operational principles relevant to the discussion, improving readability and consistency.
We also refined the scope of the review to avoid repeating biological descriptions of octopus species or sucker anatomy that have already been extensively covered in prior review articles. Instead, the revised manuscript begins from the point where biological structure meets engineering implementation that is, how octopus-inspired geometries, materials, and sensing strategies are combined in practical soft robotic systems. This shift allows us to focus on the core contribution of the manuscript: a function-oriented framework that explains how suction structures, multimodal sensors, arm/hybrid architectures, and AI-based inference are integrated to enable intelligent soft gripping. The revised figures and accompanying text now reflect this perspective directly.
We believe that these revisions address the reviewer’s concern by improving conceptual coherence, removing irrelevant details, and ensuring that the figures and narrative present a clear and focused synthesis of structural design, sensing, and control in octopus-inspired soft grippers.
- Then it would be clear why the bibliography is not in chronological order. For example, one chapter would be devoted to the implementation of suction cups, another to the implementation of tentacles, a third to the implementation of sensors on the suction cups, a fourth to the implementation of sensors for tentacle control, and a fifth chapter could be devoted to the implementation of low-level control, that is, the suction and release of the suction cups upon command from above, the flexion and extension of the tentacle, and its movement upon command from above, and a sixth chapter should be devoted to the implementation of high-level control. If the authors identify any additional problems, a separate section could also be devoted to them. If some problems remain unresolved and their solutions are not covered in the literature, then these deserve a separate discussion. In this case, it would be appropriate to indicate in the final conclusions that much remains to be solved in this area, and it would also be necessary to indicate which of the identified problems have already been sufficiently resolved and therefore do not require further in-depth research.
â–¶ We appreciate the reviewer’s thoughtful suggestion regarding how the manuscript could be organized into chapters devoted separately to suction-cup implementation, tentacle architectures, sensing on suction cups, sensing for tentacle control, low-level control mechanisms, and high-level control or AI-based strategies. We fully agree that such a component-based organization offers much clearer logic than an implicit chronological narrative, and we have substantially revised the manuscript to reflect this perspective.
In the revised version, the overall structure of the paper has been reorganized around mechanical modules, sensing architectures, and control/AI layers, rather than around timeline-based descriptions. Section 3 now focuses on the implementation of suction structures, tentacle-based designs, and their embedded sensing modules. Section 4 then addresses the operational layers, including both low-level control—such as suction engagement, release dynamics, and continuum arm actuation—and higher-level perception and AI-assisted behaviors. By restructuring the manuscript in this way, each chapter now corresponds to a clearly defined engineering layer, making the flow of concepts more coherent and more consistent with the technical evolution of the field.
This reorganization also naturally clarifies why the bibliography does not follow chronological order. Because suction mechanics, multimodal sensing, fluidic logic, and AI-driven control have developed in parallel rather than sequentially, grouping references by functional relevance rather than publication year enables a more accurate and meaningful synthesis. For the same reason, the revised manuscript begins not with biological descriptions of octopus species which have already been thoroughly addressed in previous reviews but with the point at which biological principles transition into engineering implementation. This allows us to focus directly on how structural modules, sensing systems, and control strategies interact to support intelligent soft gripping.
Finally, we have expanded the concluding section to explicitly discuss which challenges in the field remain unresolved such as long-term durability of integrated sensors, real-time multimodal fusion, and deployment in unstructured underwater environments and which problems have been sufficiently addressed by existing research. This addition responds directly to the reviewer’s suggestion that the conclusion should differentiate between well-established solutions and areas still in need of further investigation. We believe that this structural revision strengthens the clarity, coherence, and technical relevance of the review and addresses the reviewer’s concerns regarding organization, completeness, and conceptual consistency.
- A reader who has taken the time to read this article deserves to be clearly informed about which problems have been solved, which articles describe the best solutions, which problems have been solved but not effectively, and which problems remain unsolved and require urgent attention. If there are examples of studies that have addressed all of these problems, this deserves a separate mention. Instead, the reader is met with the usual generalities of bad reviews, such as how the topic of this article is highly relevant, how it is rapidly developing, how much remains to be done, and how much is currently being done. These are all unrelated and uninteresting discussions. The article doesn't feel like the authors are truly engaged in the topic of this group of studies; they exist merely as librarians who have collected some articles and compiled a summary from them. This is the level of a student's term paper, not scientific research.
â–¶ Thank you for this important comment. We agree that a reader should come away with a clear understanding of which technical issues in octopus-inspired soft grippers have been resolved, which remain only partially addressed, and which require further investigation. To address this, the revised manuscript now includes explicit analysis reflecting these distinctions.
In the revised version, Sections 3 and 4 have been restructured so that each technological dimension—structural implementation, sensing integration, low-level control, and high-level AI-assisted perception—is discussed with direct reference to the specific problems they solve and the limitations that remain. For example, Section 3 now describes suction modules, arm/hybrid structures, and embedded sensing systems not only in terms of their mechanisms but also in terms of the engineering challenges they address, such as seal stability, directional deformation sensing, or multimodal tactile perception. Section 4 then evaluates low-level actuation control (e.g., suction engagement, continuum-arm curvature regulation) and high-level inference mechanisms (e.g., ML-based material classification, pose estimation), with clear indication of where current studies provide robust solutions and where performance constraints still persist.
Most importantly, the Conclusion has been rewritten to explicitly categorize the state of the field into 1) problems for which reliable solutions exist (e.g., reversible suction, high-fidelity deformation sensing in controlled settings), 2) problems addressed only partially due to limitations in durability, environmental robustness, or computational load, and 3) unresolved challenges such as long-term sensor–material integration, real-time multimodal fusion under field conditions, and energy-efficient AI deployment.
These additions were made directly to reflect the reviewer’s request for clearer evaluation rather than general statements. The revised discussion and conclusion now identify concrete examples of effective solutions, summarize areas where current methods remain limited, and outline priority issues that require future research. We hope this revised structure provides the clarity and depth expected by readers engaging with this body of work.

Reviewer 2 Report
Comments and Suggestions for Authors
This manuscript provides a comprehensive and well-structured review of recent advances in octopus-inspired soft grippers, focusing on structural design, sensory integration, and neuromorphic intelligence. The topic is highly relevant and timely, and the overall framework is clear and logically organized. However, several sections could benefit from further clarification, elaboration, or refinement to improve the manuscript’s completeness and scholarly impact. Specific comments are as follows:
- In Table 1, the authors are advised to replace full journal citations such as “IEEE Transactions on Robotics, Vol. 40, 2024” and “Adv. Intell. Syst. 2023, 5, 2200201” with corresponding reference numbers (e.g., [50]) to improve conciseness and readability.
- The manuscript currently lacks quantitative performance evaluation and comparative analysis. It is recommended to add a summary table compiling representative studies with key quantitative indicators—such as gripper material, actuation method, sensing modality, grasping performance metrics, along with their respective advantages and limitations. Such a comparative summary would help readers quickly understand the distinctions and trade-offs among different approaches, improving both the readability and practical relevance of the review.
- On Page 3, the authors state: “By mapping structural architectures, sensing modalities, and computational strategies onto these design paradigms.” The authors are encouraged to refer to the article with doi: 10.1017/S0263574725102609 for the discussion on mapping mechanisms in soft robots design.
- Figure captions and legends should include clear reference indicators (e.g., [50]) to improve readability.
- Section 3.3, ‘Transition to Neuromorphic Integration’, explores the frontier of neuromorphic integration in soft robotics, a topic of high significance and forward-looking value. However, this section is relatively brief, and the discussion appears somewhat limited, which may not fully support readers’ understanding of its core logic. It is recommended that the authors appropriately expand this part to further elaborate on the key implementation pathways and research challenges of neuromorphic integration.
Author Response
We deeply appreciate the reviewer for their thorough reading of our manuscript and useful comments. Our item-by-item response follows:
COMMENTS TO AUTHOR:
Reviewer #2: This manuscript provides a comprehensive and well-structured review of recent advances in octopus-inspired soft grippers, focusing on structural design, sensory integration, and neuromorphic intelligence. The topic is highly relevant and timely, and the overall framework is clear and logically organized. However, several sections could benefit from further clarification, elaboration, or refinement to improve the manuscript’s completeness and scholarly impact. Specific comments are as follows:
â–¶ We sincerely thank the reviewer for the thoughtful and constructive comments. We appreciate the recognition that the manuscript provides a comprehensive and well-structured review of octopus-inspired soft grippers, including their structural design, sensory integration, and emerging AI-driven approaches. In response to the reviewer’s suggestions for further clarification and refinement, we have thoroughly revised the manuscript to strengthen conceptual continuity and improve scholarly completeness. In particular, we have clarified the functional organization of the review, expanded explanations of how suction, tentacle-based structures, and multimodal sensing modules operate as integrated systems, and refined the discussions on control frameworks and neuromorphic principles. These revisions enhance the coherence of the review and more clearly articulate how the presented framework synthesizes recent advances in design, sensing, and intelligent behavior in octopus-inspired soft grippers.
- In Table 1, the authors are advised to replace full journal citations such as “IEEE Transactions on Robotics, Vol. 40, 2024” and “Adv. Intell. Syst. 2023, 5, 2200201” with corresponding reference numbers (e.g., [50]) to improve conciseness and readability.
â–¶ We thank the reviewer for this helpful suggestion. In the revised manuscript, all full journal citations in revised Table 1 have been replaced with the corresponding reference numbers to improve clarity and consistency with the journal’s formatting style. This adjustment applies to entries such as the former “IEEE Transactions on Robotics, Vol. 40, 2024” and “Adv. Intell. Syst. 2023, 5, 2200201,” which now appear only as numbered references (e.g., [50]). We believe this modification enhances readability and aligns the table more clearly with the main reference list.
- The manuscript currently lacks quantitative performance evaluation and comparative analysis. It is recommended to add a summary table compiling representative studies with key quantitative indicators—such as gripper material, actuation method, sensing modality, grasping performance metrics, along with their respective advantages and limitations. Such a comparative summary would help readers quickly understand the distinctions and trade-offs among different approaches, improving both the readability and practical relevance of the review.
â–¶ We appreciate the reviewer’s valuable suggestion regarding the need for a quantitative performance summary. In response, the revised manuscript now includes two dedicated comparative tables that compile key quantitative indicators across representative studies. Table 1 summarizes structural and actuation-related parameters including gripper materials, actuation mechanisms, environmental compatibility, geometric scale, and maximum adhesion or grasping force allowing direct comparison of suction-based and hybrid designs under different conditions. Table 2 provides a complementary summary for sensing-integrated systems, presenting sensing modalities, functional roles, available grasping force ranges, and machine-learning–based performance metrics such as classification accuracy and regression outcomes. Together, these tables consolidate the quantitative landscape of current octopus-inspired grippers and highlight the distinctions, advantages, and limitations among different approaches. We believe these additions significantly improve the manuscript’s readability and practical relevance, as suggested by the reviewer.
- On Page 3, the authors state: “By mapping structural architectures, sensing modalities, and computational strategies onto these design paradigms.” The authors are encouraged to refer to the article with doi: 10.1017/S0263574725102609 for the discussion on mapping mechanisms in soft robots design.
â–¶ Thank you for pointing out this relevant reference. We have reviewed the article (doi: 10.1017/S0263574725102609 as reference [30]), which provides a useful discussion on mapping mechanisms and fluid-regulation strategies in soft robot design. In the revised manuscript, we have incorporated this citation in the section where we discuss the functional mapping of structural architectures, sensing modalities, and computational strategies. This reference has been added to support and contextualize the statement on design paradigm mapping, and it now appears in the updated reference list.
- Figure captions and legends should include clear reference indicators (e.g., [50]) to improve readability.
â–¶ We appreciate the reviewer’s suggestion regarding the use of consistent reference indicators in the figure captions. During the revision, the figures were reorganized and simplified to better align with the functional structure of the manuscript, and as part of this process all captions were updated to include clear reference numbers (e.g., [50]) corresponding to the cited source materials. This ensures that each adapted figure is properly attributed and improves overall readability and consistency across the manuscript.
- Section 3.3, ‘Transition to Neuromorphic Integration’, explores the frontier of neuromorphic integration in soft robotics, a topic of high significance and forward-looking value. However, this section is relatively brief, and the discussion appears somewhat limited, which may not fully support readers’ understanding of its core logic. It is recommended that the authors appropriately expand this part to further elaborate on the key implementation pathways and research challenges of neuromorphic integration.
â–¶ We appreciate the reviewer’s insightful feedback regarding previous Section 3.3 and its importance in conveying the emerging role of neuromorphic integration in soft robotic grippers. In the revised manuscript, this section has been substantially expanded and reorganized to provide a clearer and more complete explanation of the implementation pathways and technical challenges associated with neuromorphic principles. Specifically, the previous brief subsection has been integrated into a broader discussion in revised Sections 3 and 4, where sensing modules, low-level control frameworks, and AI-driven inference are now connected in a stepwise manner that naturally leads to neuromorphic architectures. This restructuring allows the neuromorphic-related content to be presented not as an isolated concept but as the culmination of structural, sensing, and operational layers established earlier in the manuscript. Additional explanations have been added to clarify event-driven sensing, distributed computation, and embodied fluidic logic, along with the practical barriers such as device-level integration, durability constraints, and processing efficiency that currently limit their deployment. We hope these revisions provide a more coherent and informative narrative that supports the reader’s understanding of how neuromorphic integration emerges from the preceding design and sensing frameworks.

Reviewer 3 Report
Comments and Suggestions for Authors
- The review does not explain how studies were selected for inclusion. Please specify: Databases searched (e.g., Scopus, Web of Science, Google Scholar); Time period covered; Inclusion/exclusion criteria; Number of papers analyzed in each design category.
- This will improve transparency and reproducibility.
- The paper provides a sequential narrative but lacks quantitative or tabular comparison of different designs.
- Consider adding: A comparison table summarizing performance metrics (e.g., adhesion force, response time, sensing range, energy efficiency); A discussion of trade-offs between structural simplicity, sensing accuracy, and computational load.
- Many examples are from Korean institutions or the same research group. While these are valuable, please ensure a balanced representation of international contributions (e.g., from groups in Italy, the US, and the UK that have led work in octopus-inspired robotics).
- Section 4 mentions some challenges but remains general. Expand with more concrete technical or practical limitations such as: Integration difficulties between soft materials and electronic sensors; Fatigue or degradation in underwater conditions; Data processing latency and energy efficiency issues in neuromorphic designs.
- Provide insights into how these challenges could be addressed in future research.
- The discussion of neuromorphic-oriented designs is currently descriptive. It would benefit from: Clear explanation of what qualifies a design as "neuromorphic" (e.g., event-driven sensing, spiking neural networks, local learning rules); Comparison with existing neuromorphic tactile systems; Clarification of the practical readiness level (TRL) of such systems.
- Figures (e.g., Figures 1–5) are informative but require clearer labeling and higher resolution.
- Please ensure: Consistent scale bars and legends; Improved figure captions explaining experimental details; Reference to all figures in the main text.
- References are inconsistently formatted (e.g., missing DOI formatting, inconsistent punctuation). Ensure conformity with Biomimetics style.
- Some older foundational works (e.g., on soft material actuation and fluidic logic) could be better cited to strengthen context.
- The text is technically rich but occasionally verbose and repetitive.
- Improve readability by shortening long sentences and reducing redundancy (especially in the abstract and introduction).
- Define all abbreviations at first appearance in the main text (e.g., CNN, VCEP, FEM).
- Verify that figure numbers match the references in the text (there are some inconsistencies between Fig. 2/3 descriptions).
- The “Funding” and “Conflict of Interest” sections are fine but ensure journal compliance with formatting.
- A “Summary Table” of key works per design stage would greatly help readers navigate the literature.

Please double check and perform a English-correction for the entire manuscript.
Author Response
We deeply appreciate the reviewer for their thorough reading of our manuscript and useful comments. Our item-by-item response follows:
COMMENTS TO AUTHOR:
Reviewer #3: The manuscript presents a comprehensive and well-organized review of octopus-inspired soft grippers, highlighting their evolution from structural mimicry to perception-integrated and neuromorphic-oriented designs. The topic is timely and relevant to the biomimetics and soft robotics community. However, before publication, a major revision is necessary to improve the rigor, clarity, and balance of the review.
â–¶ We sincerely thank the reviewer for the thorough evaluation and constructive feedback. We appreciate the reviewer’s recognition of the experimental completeness and the integration of the pneumatic soft gripper system with strain sensor-based feedback. In response to the reviewer’s insightful comments regarding the novelty and mechanistic interpretation, we have carefully revised the manuscript to provide additional clarification and supporting analysis as detailed below.
- The review does not explain how studies were selected for inclusion. Please specify: Databases searched (e.g., Scopus, Web of Science, Google Scholar); Time period covered; Inclusion/exclusion criteria; Number of papers analyzed in each design category. This will improve transparency and reproducibility.
â–¶ We appreciate the reviewer’s suggestion to clarify the literature selection and screening process. In the revised manuscript, the Introduction now includes a detailed description of how relevant studies were identified. We specify that a systematic search was conducted across Scopus, Web of Science, IEEE Xplore, and Google Scholar, using core keywords such as “octopus-inspired gripper,” “soft suction gripper,” “sensorized suction cup,” “soft tentacle actuator,” “triboelectric tactile gripper,” and “neuromorphic tactile,” together with broader terms related to soft robotics, underwater adhesion, and multimodal sensing. The primary search window covered publications from 2010 to 2025, while foundational biological studies from 1998 to 2009 were selectively included when directly relevant to suction morphology or muscular hydrostat mechanics. To ensure completeness, the reference list was expanded in the revision to incorporate additional recent studies from 2024–2025 published in leading journals such as Science Robotics, Advanced Materials, Soft Robotics, and Advanced Intelligent Systems. We also ensured balanced coverage by including representative works from major research groups in Italy, the United States, the United Kingdom, and Asia. The inclusion criteria required that studies provide implementable contributions to structural design, suction mechanics, tentacle-based actuation, multimodal or tactile sensing, AI- or ML-based perception, or neuromorphic processing, supported by FEM, CFD, or experimental validation. Purely biological morphology studies without engineering relevance were excluded. To further improve transparency, we now specify the approximate number of papers assessed in each category such as suction-based gripper structures, arm or hybrid architectures, sensing-integrated systems, and AI/neuromorphic approaches and explain that the total number of references has been increased accordingly in the revised manuscript.
- The paper provides a sequential narrative but lacks quantitative or tabular comparison of different designs. Consider adding: A comparison table summarizing performance metrics (e.g., adhesion force, response time, sensing range, energy efficiency); A discussion of trade-offs between structural simplicity, sensing accuracy, and computational load.
â–¶ Thank you for this valuable comment. In the revised manuscript, we have addressed this point by adding two quantitative comparison tables and expanding the accompanying discussion. Table 1 now summarizes key performance metrics of representative suction and hybrid gripper designs, including adhesion force, operating environment, actuation mechanism, material configuration, and relevant response characteristics. Table 2 provides a complementary comparison of sensing-integrated grippers, detailing sensing range, modality, resolution, machine-learning accuracy, and energy considerations where available. In addition to these tables, we have expanded the narrative discussion to explicitly describe the trade-offs among structural simplicity, sensing accuracy, environmental robustness, and computational load. These additions improve the clarity of comparison across design categories and support a more systematic understanding of performance distinctions and engineering constraints within octopus-inspired soft grippers.
- Many examples are from Korean institutions or the same research group. While these are valuable, please ensure a balanced representation of international contributions (e.g., from groups in Italy, the US, and the UK that have led work in octopus-inspired robotics).
â–¶ Thank you for this important comment. In the revised manuscript, we have carefully reviewed and expanded the reference list to ensure balanced representation of major international contributions. Rather than modifying the figures, we incorporated additional citations from leading groups in Italy, the United States, the United Kingdom, and Europe teams that have made foundational advances in suction mechanics, muscular-hydrostat tentacle actuation, underwater manipulation, and sensing-integrated soft grippers. These works have been integrated directly into the revised manuscript, where the structural, sensing, and control frameworks are discussed, so that the manuscript presents a more globally balanced perspective. The revision therefore emphasizes a more comprehensive and internationally representative literature base without altering the figure set, ensuring clarity and avoiding unnecessary visual modifications.
- Section 4 mentions some challenges but remains general. Expand with more concrete technical or practical limitations such as: Integration difficulties between soft materials and electronic sensors; Fatigue or degradation in underwater conditions; Data processing latency and energy efficiency issues in neuromorphic designs. Provide insights into how these challenges could be addressed in future research.
â–¶ Thank you for this helpful comment. As part of the manuscript restructuring, the original Section 4 has been reorganized into the current Section 5, and the discussion has been substantially expanded in line with the reviewer’s suggestions. The revised section now includes more concrete technical and practical limitations observed in octopus-inspired soft robotic systems. Specifically, we added detailed discussion on the integration challenges between soft materials and embedded electronic or ionic sensors, including issues related to interfacial delamination, hysteresis, and signal drift during repeated deformation. We also incorporated explicit mention of fatigue, chemical degradation, and contamination effects that arise in underwater or high-moisture environments, which remain major barriers for long-term deployment. In addition, the section now addresses data-processing latency and energy efficiency constraints in AI-assisted and neuromorphic designs, clarifying how computational load, event-driven sensing rates, and distributed processing architectures affect real-time control. We further included potential pathways for overcoming these challenges such as mechanically compliant packaging strategies, hydrophobic or antifouling coatings, hybrid fluidic–electronic sensor architectures, lightweight inference models, and hardware-level neuromorphic acceleration. These revisions strengthen the depth and specificity of the discussion and reflect the reviewer’s recommendation to provide clearer insights into the remaining obstacles and future research directions.
- The discussion of neuromorphic-oriented designs is currently descriptive. It would benefit from: Clear explanation of what qualifies a design as "neuromorphic" (e.g., event-driven sensing, spiking neural networks, local learning rules); Comparison with existing neuromorphic tactile systems; Clarification of the practical readiness level (TRL) of such systems.
â–¶ Thank you for this helpful comment. In the revised manuscript, the discussion on neuromorphic-oriented designs has been substantially reorganized and expanded within the newly structured Section 5, which replaces the previous, more limited treatment of this topic in the earlier version of the manuscript. The section title has also been updated to reflect this clarified scope. In revising this part, we refined the definition of what qualifies a design as “neuromorphic” by describing event-driven sensing behaviors, locally embedded or distributed processing, spike-inspired information encoding, and energy-efficient decision-making frameworks that distinguish these systems from conventional soft robotics signal-processing approaches. These additions were made to resolve the ambiguity of the earlier version and to provide a clearer conceptual foundation.
The revised Section 5 also includes added comparison with representative neuromorphic tactile systems reported in the broader literature, enabling readers to understand how suction-based and tentacle-based implementations relate to established neuromorphic sensing architectures such as event-based artificial skins, optical neuromorphic sensors, and spiking-inspired edge-processing modules. Additionally, we incorporated reviewer-suggested clarification on the practical readiness level of neuromorphic soft grippers, noting that most current implementations remain at an early research maturity stage with limited demonstrations outside controlled environments. The revised manuscript now outlines key developmental challenges including material–sensor co-integration, long-term operational stability, and computational efficiency that must be addressed for further progress. These updates were made to strengthen the clarity and depth of this section, in line with the reviewer’s recommendations.
- Figures (e.g., Figures 1–5) are informative but require clearer labeling and higher resolution. Please ensure: Consistent scale bars and legends; Improved figure captions explaining experimental details; Reference to all figures in the main text.
â–¶ Thank you for raising this point regarding figure clarity and formatting. In the revised manuscript, we made comprehensive updates to the figure set to address the reviewer’s suggestions. During the restructuring of Sections 3 to 5, several figures were newly added or divided into separate panels, increasing the overall number of figures and allowing each visual element to focus more clearly on a specific structural, sensing, or control concept. As part of this revision, we improved the resolution of all figures, standardized scale bars and labeling conventions, and ensured consistent use of legends to avoid ambiguity across different examples of suction modules or sensing structures.
The captions have also been rewritten to provide clearer descriptions of the mechanisms illustrated without including unnecessary information, and to better align with the newly reorganized manuscript structure. Additionally, we reviewed the main text to ensure that every figure is now explicitly referenced at the appropriate point in the narrative, reflecting the revised flow of Sections 3 to 5. These modifications were made to address the reviewer’s concerns and to improve the clarity and consistency of the visual materials in the manuscript.
- References are inconsistently formatted (e.g., missing DOI formatting, inconsistent punctuation). Ensure conformity with Biomimetics style.
â–¶ Thank you for pointing out the inconsistency in the reference formatting. In the revised manuscript, the entire reference list has been carefully reviewed and reformatted to ensure full conformity with the Biomimetics reference style. All entries now follow a consistent structure, including standardized punctuation, complete bibliographic information, and correctly formatted DOIs. We also corrected inconsistencies that appeared in the original submission such as variations in journal abbreviations, missing DOI prefixes, and uneven spacing to align the reference list with the journal’s guidelines. These adjustments were made to address the reviewer’s comment and to ensure uniformity throughout the manuscript.
- Some older foundational works (e.g., on soft material actuation and fluidic logic) could be better cited to strengthen context.
â–¶ Thank you for this helpful suggestion. In the revised manuscript, we have added several older but foundational references to strengthen the historical and conceptual context of soft material actuation and fluidic logic relevant to octopus-inspired systems. These include early studies on muscular hydrostat mechanics, soft continuum actuation principles, and fluidic logic architectures that informed later developments in suction-based and tentacle-based soft grippers. The additional citations have been incorporated into the introductory sections of the revised manuscript, where they provide clearer linkage between the biological and mechanical origins of soft actuation and the more recent implementations in sensing-integrated and neuromorphic-oriented designs. These updates were made to reflect the reviewer’s recommendation and to improve the contextual grounding of the discussion.
- The text is technically rich but occasionally verbose and repetitive. Improve readability by shortening long sentences and reducing redundancy (especially in the abstract and introduction).
â–¶ Thank you for this valuable comment. In the revised manuscript, we have shortened long sentences, removed repetitive phrasing, and clarified overly complex expressions—particularly in the abstract and introduction. The abstract has been rewritten as a single concise paragraph with shorter sentence structures and reduced redundancy, while maintaining the logical flow that reflects the structure of the review. The introduction has also been edited to streamline background information and avoid repeated descriptions of octopus-inspired features or motivation. These revisions were made to improve readability and to provide a clearer and more accessible narrative for readers.
- Define all abbreviations at first appearance in the main text (e.g., CNN, VCEP, FEM.
â–¶ Thank you for pointing out the inconsistent use of abbreviations. In the revised manuscript, all abbreviations are now defined at their first appearance in the main text, including terms such as FEM (finite element method), CFD (computational fluid dynamics), CNN (convolutional neural network), VCEP (vision-based contact event prediction), ML (machine learning), and AI (artificial intelligence). We reviewed the entire manuscript to ensure that each abbreviation follows this rule and that no undefined shorthand remains. This revision was made to improve clarity and ensure consistent readability across sections.
- Verify that figure numbers match the references in the text (there are some inconsistencies between Fig. 2/3 descriptions).
â–¶ Thank you for pointing out the inconsistencies in figure numbering. In the revised manuscript, all figure numbers and their corresponding references in the text have been checked and corrected. The descriptions of previous Figures 2 and 3, along with the surrounding paragraphs, were revised to ensure that each figure is referenced accurately and that the narrative aligns with the updated figure sequence. This consistency check was applied to all figures. These adjustments were made to avoid confusion and to ensure that each figure is clearly and correctly linked to its discussion in the main text.
- The “Funding” and “Conflict of Interest” sections are fine but ensure journal compliance with formatting.
â–¶ Thank you for this reminder. In the revised manuscript, the “Funding” and “Conflict of Interest” sections have been reviewed and formatted to comply with the Biomimetics journal guidelines. The wording, order, and formatting now follow the journal’s required structure. These sections were updated to ensure consistency with the submission standards.
- A “Summary Table” of key works per design stage would greatly help readers navigat
e the literature.
â–¶ Thank you for this helpful suggestion. In the revised manuscript, we incorporated the reviewer’s recommendation by updating the existing Table 1 and adding a new Table 2 to provide a clearer summary of key works across different design dimensions. The updated Table 1 now offers a consolidated comparison of structural and suction-based gripper designs, including material characteristics, actuation principles, operating conditions, and representative performance metrics. The newly introduced Table 2 summarizes sensing-integrated and computation-oriented systems, covering sensing modalities, functional capabilities, and available machine-learning or control-related indicators. Together, these tables serve as a summary guide that helps readers navigate the literature more effectively and understand how various structural, sensing, and computational approaches relate to one another within octopus-inspired soft gripper research.

Round 2
Reviewer 1 Report
Comments and Suggestions for Authors
This review may be published as is after editing.
No further revisions are required.
Reviewer 2 Report
Comments and Suggestions for Authors
The authors have addressed all my problems and the paper can be published now.
Reviewer 3 Report
Comments and Suggestions for Authors
The current version was addressed well and suitably.